# A Concept-Based Explainability Framework for Large Multimodal Models

**Jayneel Parekh**[1]    **Pegah Khayatan**[1]    **Mustafa Shukor**[1]

**Alasdair Newson**[1]    **Matthieu Cord**[1,2]

[1]ISIR, Sorbonne Université, Paris, France    [2]Valeo.ai, Paris, France

{jayneel.parekh, pegah.khayatan}@sorbonne-universite.fr

## Abstract

Large multimodal models (LMMs) combine unimodal encoders and large language models (LLMs) to perform multimodal tasks. Despite recent advancements towards the interpretability of these models, understanding internal representations of LMMs remains largely a mystery. In this paper, we present a novel framework for the interpretation of LMMs. We propose a dictionary learning based approach, applied to the representation of tokens. The elements of the learned dictionary correspond to our proposed concepts. We show that these concepts are well semantically grounded in both vision and text. Thus we refer to these as "multimodal concepts". We qualitatively and quantitatively evaluate the results of the learnt concepts. We show that the extracted multimodal concepts are useful to interpret representations of test samples. Finally, we evaluate the disentanglement between different concepts and the quality of grounding concepts visually and textually. Our implementation is publicly available.[1]

## 1   Introduction

Despite the exceptional capacity of deep neural networks (DNNs) to address complex learning problems, one aspect that hinders their deployment is the lack of human-comprehensible understanding of their internal computations. This directly calls into question their reliability and trustworthiness [5, 30]. Consequently, this has boosted research efforts in *interpretability/explainability* of these models i.e. devising methods to gain human-understandable insights about their decision processes. The growth in ability of DNNs has been accompanied by a similar increase in their design complexity and computational intensiveness. This is epitomized by the rise of vision transformers [11] and large-language models (LLMs) [8, 44] which can deploy up to tens of billions of parameters. The effectiveness of these models for unimodal processing tasks has spurred their use in addressing multimodal tasks. In particular, visual encoders and LLMs are frequently combined to address tasks such as image captioning and VQA [2, 25, 29, 43, 45]. This recent class of models are referred to as large multimodal models (LMMs).

For interpretability research, LMMs have largely remained unexplored. Most prior works on interpreting models that process visual data, focus on convolutional neural network (CNN) based systems and classification as the underlying task. Multimodal tasks and transformer-based architectures have both been relatively less studied. LMMs operate at the intersection of both domains. Thus, despite their rapidly growing popularity, there have been very few prior attempts at understanding representations inside an LMM [34, 35, 41, 42].

---

[1]Project page and code: `https://jayneelparekh.github.io/LMM_Concept_Explainability/`

This paper aims to bridge some of these differences and study in greater detail the intermediate representations of LMMs. To this end, motivated by the concept activation vector (CAV) based approaches for CNNs [14, 15, 17, 22], we propose a novel dictionary-learning based *Concept eXplainability* method designed for application to LMMs, titled *CoX-LMM*. Our method is used to learn a concept dictionary to understand the representations of a pretrained LMM for a given word/token of interest (Eg. 'Dog'). For this token, we build a matrix containing the LMM's internal representation of the token. We then linearly decompose this matrix using dictionary learning. The dictionary elements of our decomposition represent our concepts. The most interesting consequence of our method is that the learnt concepts exhibit a semantic structure that can be meaningfully grounded in both visual and textual domains. They are visually grounded by extracting the images which maximally activate these concepts. They can simultaneously be grounded in the textual domain by decoding the concept through the language model of the LMM and extracting the words/tokens they are most associated to. We refer to such concept representations as *multimodal concepts*. Our key contributions can be summarized as follows:

- We propose a novel concept-based explainability framework *CoX-LMM*, that can be used to understand internal representations of large multimodal models. To the best of our knowledge, this is the first effort targeting multimodal models at this scale.

- Our dictionary learning based concept extraction approach is used to extract a multimodal concept dictionary wherein each concept can be semantically grounded simultaneously in both text and vision. We also extend the previous concept dictionary-learning strategies using a Semi-NMF based optimization.

- We experimentally validate the notion of multimodal concepts through both, qualitative visualizations and quantitative evaluation. Our learnt concept dictionary is shown to possess a meaningful multimodal grounding covering diverse concepts, and is useful to locally interpret representations of test samples LMMs.

## 2 Related work

**Large Multimodal Models (LMMs)**   Large language models (LLMs) [8, 21, 33, 44] have emerged as the cornerstone of contemporary multimodal models. Typical large multimodal models (LMMs) [1, 4, 25, 26] comprise three components: LLMs, visual encoders, and light-weight connector modules to glue the two models. Remarkably, recent works have demonstrated that by keeping all pretrained models frozen and training only a few million parameters in the connector (e.g., a linear layer), LLMs can be adapted to understand images, videos, and audios [12, 23, 31, 43, 45], thus paving the way for solving multi-modal tasks. However, there is still a lack of effort aimed at understanding why such frozen LLMs can generalize to multimodal inputs. In this study, we try to decode the internal representation of LLMs when exposed to multimodal inputs.

**Concept activation vector based approaches**   Concept based interpretability aim to extract the semantic content relevant for a model [9]. For post-hoc interpretation of pretrained models, concept activation vector (CAV) based approaches [15, 17, 22, 46–48] have been most widely used. The idea of CAV was first proposed by Kim et al. [22]. They define a concept as a set of user-specified examples. The concept is represented in the activation space of deep layer of a CNN by a hyperplane that separates these examples from a set of random examples. This direction in the activation space is referred to as the concept activation vector. Built upon CAV, ACE [17] automate the concept extraction process. CRAFT [15] proposed to learn a set of concepts for a class by decomposing activations of image crops via non-negative matrix factorization (NMF). Recently, Fel et al. [14] proposed a unified view of CAV-based approaches as variants of a dictionary learning problem. However, these methods have only been applied for interpretation of CNNs on classification tasks. LMMs on the contrary exhibit a different architecture. We propose a dictionary learning based concept extraction method, designed for LMMs. We also propose a Semi-NMF variant of the dictionary learning problem, which has not been previously considered for concept extraction.

**Understanding VLM/LMM representations**   There has been an increasing interest in understanding internal representations of visual-language models (VLM) through the lens of multimodality. Shukor and Cord [42] analyse multimodal tokens and shows that despite being different, visual and perceptual tokens are implicitly aligned inside LLMs. Goh et al. [18] discover neurons termed

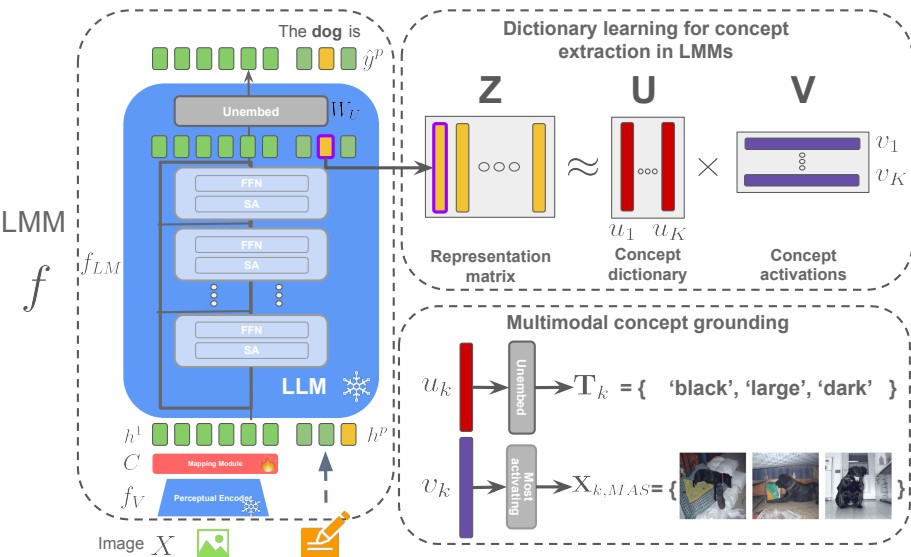

Figure 1: Overview of multimodal concept extraction and grounding in *CoX-LMM*. Given a pretrained LMM for captioning and a target token (for eg. 'Dog'), our method extracts internal representations of $f$ about $t$, across many images. These representations are collated into a matrix $\mathbf{Z}$. We linearly decompose $\mathbf{Z}$ to learn a concept dictionary $\mathbf{U}$ and its coefficients/activations $\mathbf{V}$. Each concept $u_k \in \mathbf{U}$, is multimodally grounded in both visual and textual domains. For text grounding, we compute the set of most probable words $\mathbf{T}_k$ by decoding $u_k$ through the unembedding matrix $W_U$. Visual grounding $\mathbf{X}_{k,MAS}$ is obtained via $v_k$ as the set of most activating samples.

*multimodal*, that activate for certain conceptual information given images as input. Recently proposed TEXTSPAN [16] and SpLiCE [7], aim to understand representations in CLIP [38] by decomposing its visual representations on textual representations. For LMMs, Palit et al. [34] extend the causal tracing used for LLMs to analyze information across different layers in an LMM. Schwettmann et al. [41] first proposed the notion of *multimodal neurons* existing within the LLM part of an LMM. They term the neurons "multimodal" as they translate high-level visual information to corresponding information in text modality. The neurons are discovered by ranking them by a gradient based attribution score. Pan et al. [35] proposed a more refined algorithm to identify such neurons based on a different neuron importance measure that leverages architectural information of transformer MLP blocks. Instead, we propose to discover a concept structure in the token representations by learning a small dictionary of multimodally grounded concepts. Limiting the analysis to a specific token of interest allows our method to discover fine details about the token in the learnt concepts.

## 3 Approach

### 3.1 Background for Large Multimodal Models (LMMs)

**Model architecture.** We consider a general model architecture for a large multimodal model $f$, that consists of: a visual encoder $f_V$, a trainable connector $C$, and an LLM $f_{LM}$ consisting of $N_L$ layers. We assume $f$ is pretrained for captioning task with an underlying dataset $\mathcal{S} = \{(X_i, y_i)\}_{i=1}^{N}$ consisting of images $X_i \in \mathcal{X}$ and their associated caption $y_i \subset \mathcal{Y}$. $\mathcal{X}$ and $\mathcal{Y}$ denote the space of images and set of text tokens respectively. Note that caption $y_i$ can be viewed as a subset of all tokens. The input to the language model $f_{LM}$ is denoted by the sequence of tokens $h^1, h^2, ..., h^p$ and the output as $\hat{y}$. The internal representation of any token at some layer $l$ and position $p$ inside $f_{LM}$ is denoted as $h_{(l)}^p$, with $h_{(0)}^p = h^p$. Note that $h_{(l)}^p$ is same as the residual stream representation in LLM transformers [13] at position $p$ and layer $l$. For the multimodal model, the input sequence of tokens for $f_{LM}$ consists of the concatenation of: (1) $N_V$ visual tokens provided by the visual encoder $f_V$ operating on an image $X$, followed by the connector $C$, and (2) linearly embedded textual tokens previously predicted by $f_{LM}$. For $p > N_V$, this can be expressed as:

$$\hat{y}^p = f_{LM}(h^1, h^2, \ldots, h^{N_V}, \ldots, h^p), \tag{1}$$

where $h^1, \ldots, h^{N_V} = C(f_V(X))$, and $h^p = \text{Emb}(\hat{y}^{p-1})$ for $p > N_V$, where Emb denotes the token embedding function. To start the prediction, $h^{N_V+1}$ is defined as the beginning of sentence token. The output token $\hat{y}^p$ is obtained by normalizing $h^p_{(N_L)}$, followed by an unembedding layer that applies a matrix $W_U$ followed by a softmax. The predicted caption $\hat{y}$ consists of the predicted tokens $\hat{y} = \{\hat{y}^p\}_{p > N_V}$ until the end of sentence token.

**Training** The model is trained with next token prediction objective, to generate text conditioned on images in an auto-regressive fashion. In this work we focus on models trained to "translate" images into text, or image captioning models. These models keep the visual encoder $f_V$ frozen and only train the connector $C$. Recent models also finetune the LLM to improve performance. Our approach can be applied in either case, and in our experiments we consider both type of LMMs. However, we find the generalization of LLMs to multimodal inputs is an interesting phenomenon to understand, thus we focus more on the setup where the LLM is kept frozen.

**Transformer representations view** Central to many previous approaches interpreting decoder-only LLM/transformer architectures, is the "residual stream view" of internal representations, first proposed in [13]. Herein, the network is seen as a composition of various computational blocks that "read" information from the residual stream of token representations $h^p_{(l)}$, perform their computation, and add or "write" their output back in the residual stream. This view can be summarized as:.

$$h^p_{(l+1)} = h^p_{(l)} + a^p_{(l)} + m^p_{(l)} \tag{2}$$

$a^p_{(l)}$ denotes the information computed by attention function at layer $l$ and position $p$. It has a causal structure and computes its output using $h^1_{(l)}, \ldots, h^p_{(l)}$. $m^p_{(l)}$ denotes the information computed by the MLP block. It is a feedforward network (FFN) with two fully-connected layers and an intermediate activation function $\sigma$, that operates on $h^p_{(l)} + a^p_{(l)}$. The output of $\sigma(.)$ is referred to as FFN activations.

## 3.2 Method overview

Fig. 1 provides a visual summary of the whole *CoX-LMM* pipeline. Given a pretrained LMM $f$ and a token of interest $t \in \mathcal{Y}$, our method consists of three key parts:

1. Selecting a subset of images $\mathbf{X}$ from dataset $\mathcal{S}$, relevant for target token $t$. We extract representations by processing samples in $\mathbf{X}$ through $f$. The extracted representations of dimension $B$ are collected in a matrix $\mathbf{Z} \in \mathbb{R}^{B \times M}$, where $M$ is number of samples in $\mathbf{X}$.

2. Linearly decomposing $\mathbf{Z} \approx \mathbf{UV}$ into its constituents, that includes a dictionary of learnt concepts $\mathbf{U} \in \mathbb{R}^{B \times K}$ of size $K$ and coefficient/activation matrix $\mathbf{V} \in \mathbb{R}^{K \times M}$.

3. Semantically grounding the learnt "multimodal concepts", contained in dictionary $\mathbf{U}$ in both visual and textual modalities.

We emphasize at this point that our main objective in employing dictionary learning based concept extraction is to understand internal representations of an LMM. Thus, our focus is on validating the use of the learnt dictionary for this goal, and not to interpret the output of the model, which can be readily accomplished by combining this pipeline with some concept importance estimation method [14]. The rest of the section is devoted to elaborate on each of the above three steps.

## 3.3 Representation extraction

To extract relevant representations from the LMM about $t$ that encode meaningful semantic information, we first determine a set of samples $\mathbf{X}$ from dataset $\mathcal{S} = \{(X_i, y_i)\}_{i=1}^N$ for extraction. We consider the set of samples where $t$ is predicted as part of the predicted caption $\hat{y}$. This allows us to further investigate the model's internal representations of $t$. To enhance visual interpretability for the extracted concept dictionary, we additionally limit this set of samples to those that contain $t$ in the ground-truth caption. Thus, $\mathbf{X}$ is computed as:

$$\mathbf{X} = \{X_i \mid t \in f(X_i) \text{ and } t \in y_i \text{ and } (X_i, y_i) \in \mathcal{S}\}. \tag{3}$$

Given any $X \in \mathbf{X}$, we propose to extract the residual stream representation $h^p_{(L)}$ from a deep layer $L$, at the first position in the predicted caption $p > N_V$, such that $\hat{y}^p = t$. The representation $z_j \in \mathbb{R}^B$

of each sample $X_j \in \mathbf{X}$ is then stacked as columns of the matrix $\mathbf{Z} = [z_1, ..., z_M] \in \mathbb{R}^{B \times M}$. Note that the representations of text tokens in $f_{LM}$ can possess a meaningful multimodal structure as they combine information from visual token representations $h_{(l)}^p, p \leq N_V$. In contrast to $a_{(l)}^p$ and $m_{(l)}^p$, that represent residual information at layer $l$, $h_{(L)}^p$ contains the aggregated information computed by the LMM till layer $L$, providing a holistic view of its computation across all previous layers.

## 3.4 Decomposing the representations

The representation matrix $\mathbf{Z} \approx \mathbf{UV}$, is decomposed as product of two low-rank matrices $\mathbf{U} \in \mathbb{R}^{B \times K}, \mathbf{V} \in \mathbb{R}^{K \times M}$ of rank $K << \min(B, M)$, where $K$ denotes the number of dictionary elements. The columns of $\mathbf{U} = [u_1, ..., u_K]$ are the basis vectors which we refer to as concept-vectors/concepts. The rows of $\mathbf{V}$ or columns of $\mathbf{V}^T = [v_1, ..., v_K], v_i \in \mathbb{R}^M$ denote the activations of $u_i$ for each sample. This decomposition, as previously studied in [14], can be optimized with various constraints on $\mathbf{U}, \mathbf{V}$, each leading to a different dictionary. The most common ones include PCA (constraint: $\mathbf{U}^T \mathbf{U} = \mathbf{I}$), K-Means (constraint: columns of $\mathbf{V}$ correspond to columns of identity matrix) and NMF (constraint: $\mathbf{U}, \mathbf{V} \geq 0$). NMF is considered to yield most interpretable results [14]. However, for our use case, NMF is not useful as representation matrix $\mathbf{Z}$ is not non-negative. Instead, we propose to employ a relaxed version of NMF, Semi-NMF [10], which allows the decomposition matrix $\mathbf{Z}$ and basis vectors $\mathbf{U}$ to contain mixed values, and only forces the activations $\mathbf{V}$ to be non-negative. Note that given its relations to clustering algorithms [10], enforcing non-negative combinations of decompositions is still valued from an interpretability perspective. Since we expect only a small number of concepts to be present in any given sample, we also encourage sparsity in activations $\mathbf{V}$. The optimization problem to decompose $\mathbf{Z}$ via Semi-NMF can be summarized as:

$$\mathbf{U}^*, \mathbf{V}^* = \arg\min_{\mathbf{U}, \mathbf{V}} \ ||\mathbf{Z} - \mathbf{UV}||_F^2 + \lambda ||\mathbf{V}||_1 \quad s.t. \ \mathbf{V} \geq 0, \text{ and } ||u_k||_2 \leq 1 \ \forall k \in \{1, ..., K\}. \quad (4)$$

Given any image $X$ where token $t$ is predicted by $f$, we can now define the process of computing activations of concept dictionary $\mathbf{U}^*$ for given $X$, denoted as $v(X) \in \mathbb{R}^K$. To do so, we first extract the token representation for $X, z_X \in \mathbb{R}^B$ with the process described in Sec. 3.3. Then, $z_X$ is projected on $\mathbf{U}^*$ to compute $v(X)$. In the case of Semi-NMF, this corresponds to $v(X) = \arg\min_{v \geq 0} ||z_X - \mathbf{U}^* v||_2^2 + \lambda ||v||_1$. The activation of $u_k \in \mathbf{U}^*$ is denoted as $v_k(X) \in \mathbb{R}$.

## 3.5 Using the concept dictionary for interpretation

**Multimodal grounding of concepts.** Given the learnt dictionary $\mathbf{U}^*$ and corresponding activations $\mathbf{V}^*$, the key objective remaining is to ground the understanding of any given concept vector $u_k, k \in \{1, ..., K\}$ in the visual and textual domains. Specifically, for visual grounding, we use prototyping [3, 22] to select input images (among the decomposed samples), that maximally activate $u_k$. Given the number of samples extracted for visualization $N_{MAS}$, the set of maximum activating samples (MAS) for component $u_k$, denoted as $\mathbf{X}_{k,MAS}$ can be specified as follows ($|.|$ is absolute value):

$$\mathbf{X}_{k,MAS} = \operatorname*{argmax}_{\hat{X} \subset \mathbf{X}, |\hat{X}| = N_{MAS}} \sum_{X \in \hat{X}} |v_k(X)|. \quad (5)$$

For grounding in textual domain, we note that the concept vectors are defined in the token representation space of $f_{LM}$. Thus we leverage the insights from "Lens" based methods [6, 24, 32, 40] that attempt to understand LLM representations. In particular, following [32], we use the unembedding layer to map $u_k$ to the token vocabulary space $\mathcal{Y}$, and extract the most probable tokens. That is, we extract the tokens with highest probability in $W_U u_k$. The decoded tokens with highest probabilities are then filtered for being an english, non-stop-word with at least 3 characters, to eliminate unnecessary tokens. The final set of words is referred to as grounded words for concept $u_k$ and denoted as $\mathbf{T}_k$. Fig. 2 illustrates an example of grounding of a concept extracted for token "Dog" in vision (5 most activating samples) and text (top 5 decoded words).

**Most activating concepts for images.** To understand the LMM's representation of a given image $X$, we now define the *most activating concepts*. Firstly, we extract the representaions $z_X$ of the image with the same process as described previously. We then project $z_X$ on $\mathbf{U}^*$ to obtain $v(X) \in \mathbb{R}^K$. We define the most activating concepts, which we denote $\tilde{u}(X)$, as the set of $r$ concept vectors (in

'white'
'light'
'fluffy'
'golden'
'dog'

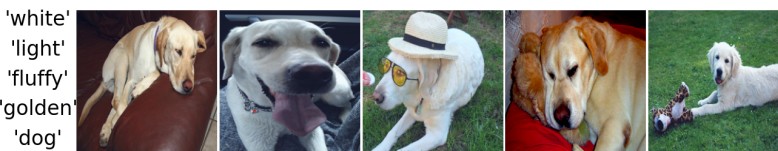

Figure 2: Example of multimodal concept grounding in vision and text. Five most activating samples (among decomposed in $\mathbf{Z}$) and five most probable decoded words are shown.

$\mathbf{U}^*$) whose activations $v_k(X)$ have the largest magnitude. One can then visualize the multimodal grounding of $\tilde{u}(X)$. This step could be further combined with concept importance estimation techniques [14] to interpret the model's prediction for token $t$, however, the focus of this paper is to simply understand the internal representation of the model, for which the current pipeline suffices.

## 4 Experiments

**Models and dictionary learning.** In the main paper, we focus on experiments with the DePALM model [45] that is trained for captioning task on COCO dataset [27]. The model consists of a frozen ViT-L/14 CLIP [39] encoder as the visual encoder $f_V$. It is followed by a transformer connector to compress the encoding into $N_V = 10$ visual tokens. The language model $f_{LM}$ is a frozen OPT-6.7B [49] and consists of 32 layers. Additional experiments with LLaVA [28] are in Appendix A. For uniformity and fairness all the results in the main paper are reported with number of concepts $K = 20$ and for token representations from $L = 31$, the final layer before unembedding layer. For Semi-NMF, we set $\lambda = 1$ throughout. We consider the 5 most activating samples in $\mathbf{X}_{k,MAS}$ for visual grounding for any $u_k$. For text grounding, we consider top-15 tokens for $\mathbf{T}_k$ before applying the filtering described in Sec 3.5.

The complete dataset consists of around 120,000 images for training, and 5000 each for validation and testing with 5 captions per image following the Karpathy split. We conduct our analysis separately for various common objects in the dataset: "Dog", "Bus", "Train", "Cat", "Bear", "Baby", "Car", "Cake". The extension to other classes/tokens remains straightforward and is discussed in Appendix D. The precise details about number of samples for learning the dictionary, or testing, is available in Appendix C. The implementation of our method is publicly available on GitHub [2]

### 4.1 Evaluation setup

We evaluate the quality of learnt concept dictionary $\mathbf{U}^*$ on three axes: (i) Its use during inference to interpret representations of LMMs for test samples, (ii) The overlap/entanglement between grounded words of concepts in the dictionary and (iii) the quality of visual and text grounding of concepts (used to understand a concept itself). We discuss concrete details about each axis below:

**Concept extraction during inference:** To evaluate the use of $\mathbf{U}^*$ in understanding any test sample $X$, we first estimate the top $r$ most activating concepts activations, $\tilde{u}(X)$ (Sec. 3.5). We then estimate the correspondence between the test image $X$ and the grounded words $\mathbf{T}_k$ of $u_k \in \tilde{u}(X)$. This correspondence is estimated via two different metrics. The primary metric is the average CLIPScore [20] between $X$ and $\mathbf{T}_k$. This directly estimates correspondence between the test image embedding with the grounded words of the top concepts. The secondary metric is the average BERTScore (F1) [50] between the ground-truth captions $y$ associated with $X$ and the words $\mathbf{T}_k$. These metrics help validate the multimodal nature of the concept dictionaries. Their use is inspired from [41]. Details for their implementation is in Appendix C.

**Overlap/entanglement of learnt concepts:** Ideally, we would like each concept in $\mathbf{U}^*$ to encode distinct information about the token of interest $t$. Thus two different concepts $u_k, u_l, k \neq l$ should be associated to different sets of words. To quantify the entanglement of learnt concepts, we compute the overlap between the grounded words $\mathbf{T}_k, \mathbf{T}_l$. The overlap for a concept $u_k$ is defined as an average of its fraction of common words with other concepts. The overlap/entanglement metric for a dictionary

---

[2]`https://github.com/mshukor/xl-vlms`

| Token | Metric | Rnd-Words | Noise-Imgs | Simple | Semi-NMF (Ours) | GT-captions |
|-------|--------|-----------|------------|--------|-----------------|-------------|
| Dog | CS top-1 ($\uparrow$) | $0.519 \pm 0.05$ | $0.425 \pm 0.06$ | $\underline{0.546 \pm 0.08}$ | $\mathbf{0.610 \pm 0.09}$ | $0.783 \pm 0.06$ |
|     | BS top-1 ($\uparrow$) | $0.201 \pm 0.04$ | $0.306 \pm 0.05$ | $\underline{0.346 \pm 0.08}$ | $\mathbf{0.405 \pm 0.07}$ | $0.511 \pm 0.11$ |
| Bus | CS top-1 ($\uparrow$) | $0.507 \pm 0.05$ | $0.425 \pm 0.08$ | $\mathbf{0.667 \pm 0.06}$ | $\underline{0.634 \pm 0.08}$ | $0.736 \pm 0.05$ |
|     | BS top-1 ($\uparrow$) | $0.200 \pm 0.05$ | $0.303 \pm 0.06$ | $\underline{0.390 \pm 0.05}$ | $\mathbf{0.404 \pm 0.07}$ | $0.466 \pm 0.11$ |
| Train | CS top-1 ($\uparrow$) | $0.496 \pm 0.05$ | $0.410 \pm 0.07$ | $\underline{0.642 \pm 0.06}$ | $\mathbf{0.646 \pm 0.07}$ | $0.727 \pm 0.05$ |
|       | BS top-1 ($\uparrow$) | $0.210 \pm 0.06$ | $0.253 \pm 0.06$ | $\mathbf{0.392 \pm 0.07}$ | $\underline{0.378 \pm 0.07}$ | $0.436 \pm 0.08$ |
| Cat | CS top-1 ($\uparrow$) | $0.539 \pm 0.04$ | $0.461 \pm 0.04$ | $\underline{0.589 \pm 0.07}$ | $\mathbf{0.627 \pm 0.06}$ | $0.798 \pm 0.05$ |
|     | BS top-1 ($\uparrow$) | $0.207 \pm 0.07$ | $0.307 \pm 0.03$ | $\underline{0.425 \pm 0.10}$ | $\mathbf{0.437 \pm 0.08}$ | $0.544 \pm 0.10$ |

Table 1: Test data mean CLIPScore and BERTScore for top-1 activating concept for all baselines on five tokens. CLIPScore denoted as CS, and BERTScore as BS. Statistical significance is in Appendix D. Our *CoX-LMM* framework is evaluated with Semi-NMF as underlying dictionary learning method. Higher scores are better. Best score in **bold**, second best is underlined.

$\mathbf{U}^*$ is defined as the average of overlap of each concept.

$$\text{Overlap}(\mathbf{U}^*) = \frac{1}{K} \sum_k \text{Overlap}(u_k), \quad \text{Overlap}(u_k) = \frac{1}{(K-1)} \sum_{l=1, l \neq k}^{K} \frac{|\mathbf{T}_l \cap \mathbf{T}_k|}{|\mathbf{T}_k|}$$

**Multimodal grounding of concepts:** To evaluate the quality of visual/text grounding of concepts $(\mathbf{X}_{k,MAS}, \mathbf{T}_k)$, we measure the correspondence between visual and text grounding of a given concept $u_k$, i.e. the set of maximum activating samples $\mathbf{X}_{k,MAS}$ and words $\mathbf{T}_k$, using CLIPScore and BERTScore as described above.

**Baselines:** One set of methods for evaluation are the variants of *CoX-LMM* where we employ different dictionary learning strategies: PCA, KMeans and Semi-NMF. For evaluating concept extraction on test data with CLIPScore/BERTScore we compare against the following baselines:
- *Rnd-Words*: This baseline considers Semi-NMF as the underlying learning method. However, for each component $u_k$, we replace its grounded words $\mathbf{T}_k$ by a set of random words $\mathbf{R}_k$ such that $|\mathbf{R}_k| = |\mathbf{T}_k|$ and the random words also satisfy the same filtering conditions as grounded words i.e. they are non-stopwords from english corpus with more than two characters. We do this by decoding a randomly sampled token representation and adding the top decoded words if they satisfy the conditions.
- *Noise-Imgs*: This baseline uses random noise as images and then proceeds with exactly same learning procedure as Semi-NMF including extracting activations from the same positions, and same parameters for dictionary learning. Combined with the Rnd-Words baseline, they ablate two parts of the concept extraction pipeline.
- *Simple*: This baseline considers a simple technique to build the dictionary $\mathbf{U}^*$ and projecting test samples. It builds $\mathbf{U}^*$ by selecting token representations in $\mathbf{Z}$ with the largest norm. The projections are performed by mapping the test sample representation to the closest element in $\mathbf{U}^*$. For deeper layers, this provides a strong baseline in terms of extracted grounded words $\mathbf{T}_k$ which are related to token of interest $t$, as they are obtained by directly decoding token representations of $t$.
We also report score using ground-truth captions *(GT captions)* instead of grounded words $\mathbf{T}_k$, to get the best possible correspondence score. The overlap/entanglement in concept dictionary is compared between the non-random baselines: Simple, PCA, K-Means and Semi-NMF. For evaluating the visual/text grounding we compare against the random words keeping the underlying set of MAS, $\mathbf{X}_{k,MAS}$, same for both.

## 4.2   Results and discussion

**Quantitative results**     Tab. 1 reports the test top-1 CLIPScore/BERTScore for all baselines and Semi-NMF on different target tokens. Appendix D contains detailed results for other tokens as well as for the PCA and K-Means variants. We report the results for only the top-1 activating concept, as the KMeans and Simple baselines map a given representation to a single cluster/element.

Notably, Semi-NMF generally outperforms the other baselines although the Simple baseline performs competitively. More generally, Semi-NMF, K-Means, and Simple tend to clearly outperform Rnd-Words, Noise-Imgs and PCA on these metrics, indicating that these systems project representations of test images to concepts whose associated grounded words correspond well with the visual content.

Tab. 2 reports the overlap between concepts for Simple baseline and PCA, K-Means and Semi-NMF variants of *CoX-LMM*. Interestingly, K-Means and Simple baseline perform significantly worse than Semi-NMF/PCA with a high overlap between grounded words, often exceeding 40%. PCA outperforms other methods with almost no overlap while Semi-NMF shows some overlap. Overall, Semi-NMF strikes the best balance among all the methods, in terms of learning a concept dictionary useful for understanding test image representations, but which also learns diverse and disentangled concepts. Thus, for further *CoX-LMM* experiments, we consider Semi-NMF as the underlying dictionary learning method.

| Token | Simple | PCA | KMeans | Semi-NMF |
|-------|--------|-------|--------|----------|
| Dog | 0.371 | **0.004** | 0.501 | 0.086 |
| Bus | 0.622 | **0.002** | 0.487 | 0.177 |
| Train | 0.619 | **0.015** | 0.367 | 0.107 |
| Cat | 0.452 | **0.000** | 0.500 | 0.146 |

Table 2: Overlap between learnt concepts. Lower is better. Best score in **bold**, second best underlined.

Fig. 3 shows an evaluation of visual/text grounding of concepts learnt by Semi-NMF. Each point on the figure denotes the CLIP-Score (left) or BERTScore (right) for correspondence between samples $\mathbf{X}_{k,MAS}$ and words $\mathbf{T}_k$ for concept $u_k$ against random words baseline. We see that for both metrics, vast majority of concepts lie above the $y = x$ line, indicating that grounded words correspond much better to content of maximum activating samples than random words.

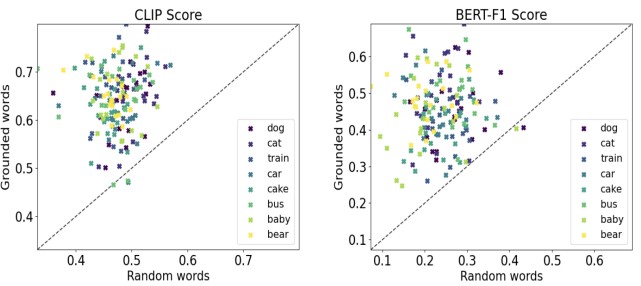

Figure 3: Evaluating visual/text grounding (CLIP-Score/BERTScore). Each point denotes score for grounded words of a concept (Semi-NMF) vs Rnd-Words w.r.t the same visual grounding.

**Qualitative results** Fig. 4 shows visual and textual grounding of concepts extracted for token 'dog'. For brevity, we select 8 out of 20 concepts for illustration. 2. Grounding for all concepts extracted for 'dog' and other tokens are in Appendix E. The concept visualizations/grounding for

**Example concepts for 'Dog'**

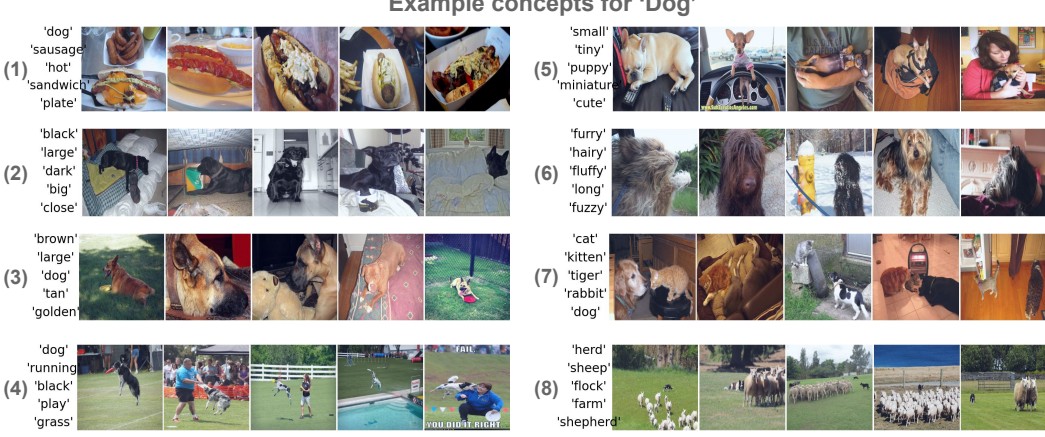

Figure 4: Visual/textual grounding for 8 out of 20 concepts for 'Dog' token (layer 31). For each concept we illustrate the set of 5 most activating samples and 5 most probable decoded words.

'Dog' reveal interesting insights about the global structure of the LMM's representation. Extracted concepts capture information about different aspects of a 'dog'. The LMM separates representation

of animal 'Dog' with a 'hot dog' (Concept 1). Specifically for 'Dog', Concepts (2), (3) capture information about color: 'black', 'brown'. Concept (6) encodes information about 'long hair' of a dog, while concept (5) activates for 'small/puppy-like' dogs. Beyond concepts activating for specific characteristics of a 'dog', we also discover concepts describing their state of actions (Concept (4) 'playing/running'), common scenes they can occur in (Concept (8), 'herd'), and correlated objects they can occur with (Concept (7), 'cat and dog'). We observe such diverse nature of extracted concepts even for other tokens (Appendix E). The information about concepts can be inferred via both the visual images and the associated grounded words, highlighting their coherent multimodal grounding. Notably, compared to solely visual grounding as for CAVs for CNNs, the multimodal grounding eases the process to understand a concept.

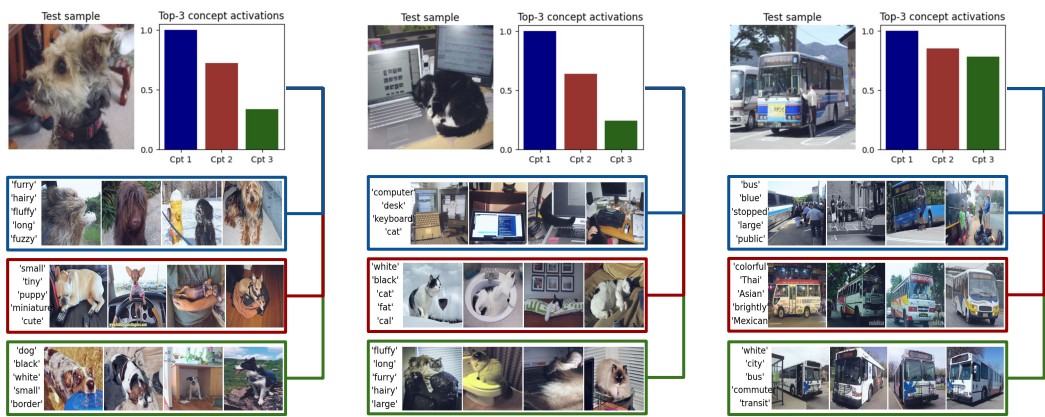

Figure 5: Local interpretations for test samples for different tokens ('Dog', 'Cat', 'Bus') with Semi-NMF (layer 31). Visual/text grounding for three highest concept activations (normalized) is shown.

Fig. 5 illustrates the use of concept dictionaries (learnt via Semi-NMF) to understand test sample representations for tokens 'Dog', 'Cat' and 'Bus'. For each sample we show the normalized activations of the three most activating concepts, and their respective multimodal grounding. Most activating concepts often capture meaningful and diverse features of a given sample. For instance, for first sample containing a 'Dog', the concepts for "long hair", "small/tiny/puppy", and "black/white color" all simultaneously activate. The grounding for first two concepts was also illustrated in Fig. 4. Additional visualizations for test samples are shown in Appendix E, wherein we qualitatively compare interpretations of Semi-NMF to K-Means, PCA variants and Simple baseline.

**Layer ablation** We analyze the quality of multimodal grounding of concepts across different layers $L$. The CLIPScore between $(\mathbf{X}_{k,MAS}, \mathbf{T}_k)$, averaged over all concepts $u_k$ is shown in Fig. 6, for 'Dog' and 'Cat' for all layers in $f_{LM}$. For early layers the multimodal grounding is no better than Rnd-Words. Interestingly, there is a noticeable increase around ($L = 20$ to $L = 25$), indicating that the multimodal structure of internal token representations starts to appear at this point. This also validates our choice that deeper layers are better suited for multimodal concepts.

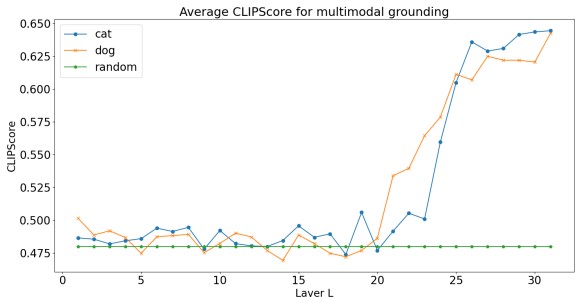

Figure 6: Mean CLIPScore between visual/text grounding $\mathbf{X}_{k,MAS}, \mathbf{T}_k$ for all concepts (Semi-NMF), across different layers $L$. Results are for tokens 'Dog' and 'Cat'.

**Additional experiments and discussion.** We conduct a preliminary study to analyze the polysemanticity/superposition in concept dictionaries in Appendix B. A qualitative analysis for grounding of extracted concepts for different layers is available in Appendix F. *CoX-LMM* can be also be applied to understand the processing of visual/perceptual tokens inside the LMM which also exhibit this multimodal structure. The experiment for the same can be found in Appendix G. Limitations of our method are discussed in Appendix H, and the broader societal impacts are discussed in Appendix I.

# 5 Conclusion

In summary, we have presented a novel dictionary learning based concept extraction framework, useful to understand internal representations of a large multimodal model. The approach relies on decomposing representations of a token inside a pretrained LMM. To this end, we also propose a Semi-NMF variant of the concept dictionary learning problem. The elements of the learnt concept dictionary are grounded in the both text and visual domains, leading to a novel notion of *multimodal concepts* in the context of interpretability. We quantitatively and qualitatively show that (i) the multimodal grounding of concepts is meaningful, and (ii) the learnt concepts are useful to understand representations of test samples. We hope that our method inspires future work from research community towards designing concept based explainability methods to understand LMMs.

## Acknowledgments and Disclosure of Funding

We would like to thank Thomas Fel for his valuable comments on the paper. This work has been partially supported by ANR grant VISA DEEP (ANR-20-CHIA-0022), and HPC resources of IDRIS under the file number AD011014947, allocated by GENCI.

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

# A  Experiments on other LMMs

This section covers our experiments on other types of multimodal models. First, we test our approach on LLaVA to demonstrate that our approach generalizes to more recent networks that also fine-tune $f_{LM}$ on multimodal data. We also test our method on other variants of DePALM with non-CLIP visual encoders to observe their effect on CLIPScore.

## A.1  Experiments with LLaVA

We conduct further experiments on LLaVA [28], a popular open-source LMM to demonstrate the generality of our method. The model uses a CLIP-ViT-L-336px visual encoder ($f_V$), a 2-layer linear connector ($C$) that outputs $N_V = 576$ visual tokens, and a Vicuna-7B language model ($f_{LM}$, 32 layers). We use identical hyperparameters as for DePALM ($K = 20, \lambda = 1, L = 31$). We report the test CLIPScore for top-1 activating concept, for Rnd-Words, Noise-Imgs, Simple and Semi-NMF, GT-captions in Tab. 3, and Overlap score for non-random baselines in Tab. 4. Quantitatively, we obtain consistent results to those observed for DePALM. Semi-NMF extracts most balanced concept dictionaries with high multimodal correspondence and low overlap. Qualitatively too, the method functions consistently and is able to extract concepts with meaningful multimodal grounding.

| Token | Metric | Rnd-Words | Noise-Imgs | Simple | Semi-NMF (Ours) | GT-captions |
|---|---|---|---|---|---|---|
| Dog | CS top-1 (↑) | $0.537 \pm 0.03$ | $0.530 \pm 0.05$ | $\underline{0.567 \pm 0.08}$ | $\mathbf{0.595 \pm 0.07}$ | $0.777 \pm 0.06$ |
|  | BS top-1 (↑) | $0.205 \pm 0.07$ | $0.227 \pm 0.06$ | $\mathbf{0.331 \pm 0.07}$ | $\underline{0.305 \pm 0.07}$ | $0.519 \pm 0.11$ |
| Bus | CS top-1 (↑) | $0.509 \pm 0.04$ | $0.487 \pm 0.05$ | $\mathbf{0.619 \pm 0.06}$ | $\underline{0.591 \pm 0.08}$ | $0.742 \pm 0.05$ |
|  | BS top-1 (↑) | $0.198 \pm 0.07$ | $0.253 \pm 0.06$ | $\mathbf{0.319 \pm 0.04}$ | $\underline{0.306 \pm 0.06}$ | $0.460 \pm 0.10$ |
| Train | CS top-1 (↑) | $0.518 \pm 0.03$ | $0.505 \pm 0.04$ | $\underline{0.633 \pm 0.05}$ | $\mathbf{0.640 \pm 0.07}$ | $0.725 \pm 0.05$ |
|  | BS top-1 (↑) | $0.177 \pm 0.07$ | $0.221 \pm 0.04$ | $\mathbf{0.310 \pm 0.05}$ | $\underline{0.293 \pm 0.05}$ | $0.432 \pm 0.08$ |
| Cat | CS top-1 (↑) | $0.536 \pm 0.03$ | $0.545 \pm 0.04$ | $\mathbf{0.625 \pm 0.06}$ | $\underline{0.621 \pm 0.07}$ | $0.795 \pm 0.05$ |
|  | BS top-1 (↑) | $0.142 \pm 0.06$ | $0.235 \pm 0.05$ | $\underline{0.306 \pm 0.06}$ | $\mathbf{0.329 \pm 0.07}$ | $0.540 \pm 0.11$ |

Table 3: Concept extraction on LLaVA-v1.5: Test data mean CLIPScore reported for top-1 activating concept for same baselines and tokens as in main paper table 1. Higher scores are better. Best score in **bold**, second best is underlined.

| Token | Simple | PCA | KMeans | Semi-NMF |
|---|---|---|---|---|
| Dog | 0.435 | **0.008** | 0.429 | 0.149 |
| Bus | 0.464 | **0.010** | 0.518 | 0.124 |
| Train | 0.315 | **0.024** | 0.382 | 0.087 |
| Cat | 0.479 | **0.013** | 0.554 | 0.166 |

Table 4: Overlap evaluation (LLaVA). Lower is better. Best score in **bold**, second best underlined.

**Qualitative results and Saliency maps**  We also show qualitative examples of concepts extracted for token 'Dog' in Fig. 7. More examples for other 'Cat' and 'Train' tokens are given in Fig. 9 and 10. Interestingly, since LLaVA uses a connector $C$ that contains two linear layers, the visual tokens as processed inside $f_{LM}$ preserve the notion of specific image patch representations, i.e. $N_V = 576$ tokens denoting representations for 576 ($24 \times 24$) input patches. This allows us to further explore a simple and computationally cheap strategy of generating saliency maps to highlight which regions a concept vector activates on. To do this one can simply compute the inner product of any given concept vector $u_k$ with all visual token representations from corresponding layer $L$, i.e. $u_k^T [h_{(L)}^1, ..., h_{(L)}^{N_V}]$. This can be upscaled to the input image size to visualize the saliency map. We illustrate some qualitative outputs on concepts from 'Dog' in Fig. 8. .

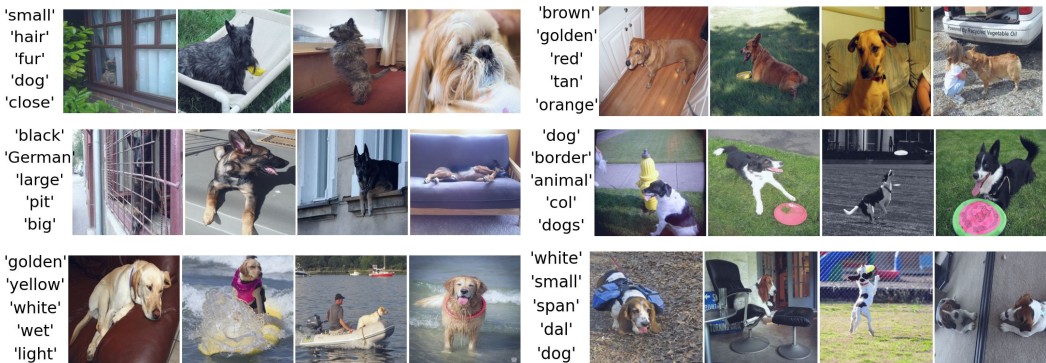

Figure 7: Multimodal grounding for example concepts for 'Dog' token (layer 31) on LLaVA.

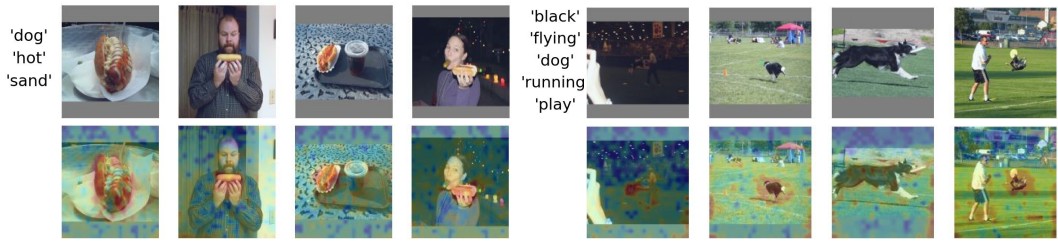

Figure 8: Examples of generating visual concept saliency maps for two 'Dog' concepts for LLaVA. Red denotes high activations, blue denotes low activation (bottom row)

## A.2   DePALM with ViT visual encoders

We further test LMMs which do not contain a CLIP visual encoder to confirm that the high CLIPScore is not due to use of CLIP visual encoders. To test this, we experiment on two different DePALM models with frozen visual encoders different from CLIP, a frozen ViT-L encoder trained on ImageNet [11] and another frozen ViT-L trained as a masked autoencoder (MAE) [19]. Both LMMs use the same pretrained OPT-6.7B language model. Collectively, the three encoders (including CLIP) are pretrained for three different types of objectives. We use Semi-NMF to extract concept dictionaries, with all hyperparameters identical. The results are reported in tables below. 'Rnd-Words' and 'GT-captions' references are reported for each LMM separately, although they are very close to the ones in main paper. The "ViT-L (CLIP)" baseline denotes our system from the main paper that uses CLIP encoder. Importantly, we still obtain similar test CLIPScores as with CLIP visual encoder. The concept dictionaries still possess meaningful multimodal grounding. Many concepts also tend to be similar as for CLIP visual encoder, further indicating that processing inside language model plays a major role in the discovery of multimodally grounded concepts.

| Token | Rnd-Words | ViT-L (ImageNet) | ViT-L (CLIP) | GT-captions |
|-------|-----------|------------------|--------------|-------------|
| Dog   | $0.514 \pm 0.05$ | $0.611 \pm 0.09$ | $0.610 \pm 0.09$ | $0.783 \pm 0.06$ |
| Bus   | $0.498 \pm 0.05$ | $0.644 \pm 0.07$ | $0.634 \pm 0.08$ | $0.739 \pm 0.05$ |
| Train | $0.494 \pm 0.05$ | $0.617 \pm 0.07$ | $0.646 \pm 0.07$ | $0.728 \pm 0.05$ |
| Cat   | $0.539 \pm 0.05$ | $0.628 \pm 0.07$ | $0.627 \pm 0.06$ | $0.794 \pm 0.06$ |

Table 5: Test CLIPScore evaluation for DePALM with ViT-L frozen image encoder trained on ImageNet: Scores reported for top-1 activating concept of Semi-NMF for Rnd-Words, GT-captions and ViT-L (CLIP) which denotes the system in main text.

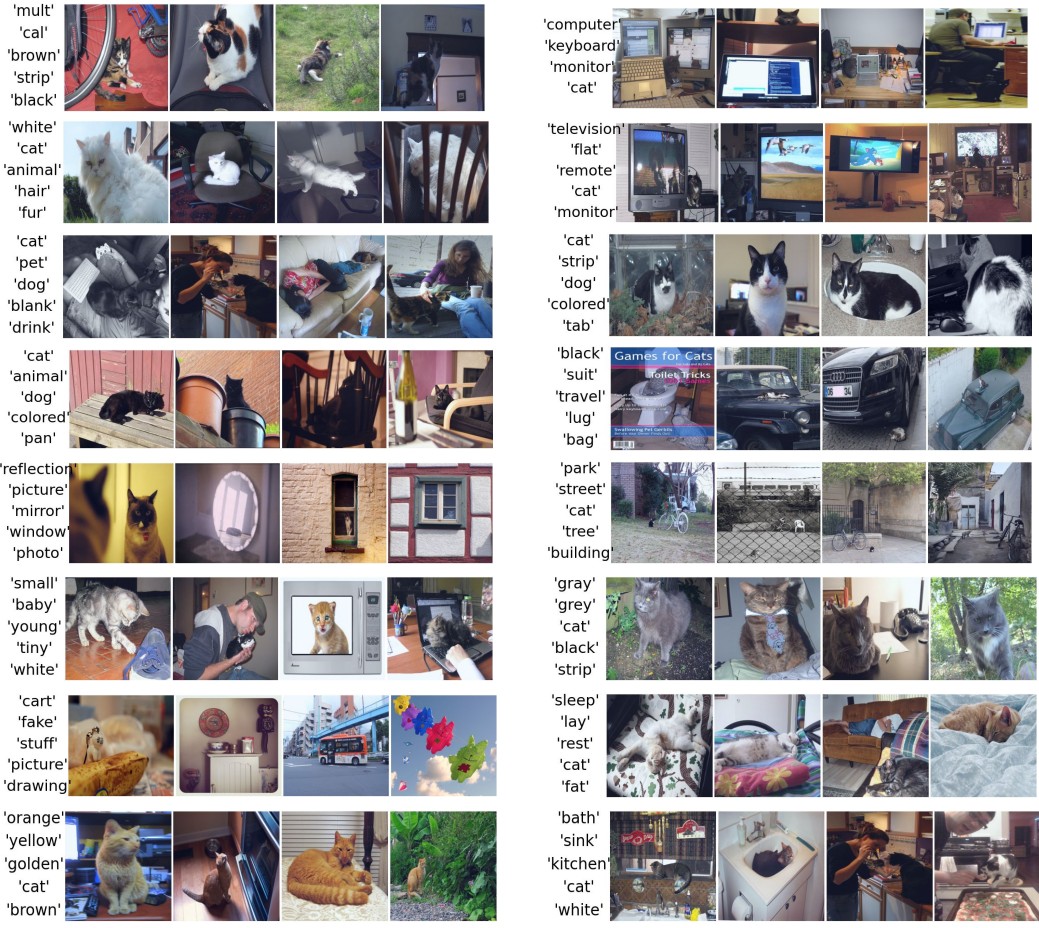

Figure 9: Multimodal grounding for example concepts for 'Cat' token (layer 31) on LLaVA.

| Token | Rnd-Words | ViT-L (MAE) | ViT-L (CLIP) | GT-captions |
|-------|-----------|-------------|--------------|-------------|
| Dog   | $0.515 \pm 0.05$ | $0.602 \pm 0.07$ | $0.610 \pm 0.09$ | $0.784 \pm 0.06$ |
| Bus   | $0.501 \pm 0.05$ | $0.627 \pm 0.07$ | $0.634 \pm 0.08$ | $0.737 \pm 0.05$ |
| Train | $0.483 \pm 0.06$ | $0.618 \pm 0.08$ | $0.646 \pm 0.07$ | $0.726 \pm 0.05$ |
| Cat   | $0.541 \pm 0.04$ | $0.629 \pm 0.09$ | $0.627 \pm 0.06$ | $0.795 \pm 0.06$ |

Table 6: Test CLIPScore evaluation for DePALM with ViT-L frozen image encoder trained as masked autoencoder (MAE): Scores reported for top-1 activating concept of Semi-NMF for Rnd-Words, GT-captions and ViT-L (CLIP) which denotes the system in main text.

## B  Analyzing polysemanticity in the learnt concepts

We conducted a preliminary qualitative study on some concept vectors in the dictionary learnt for token "Dog" (DePALM model), to analyze if these concept vectors tend to activate strongly for a specific semantic concept (monosemantic) or multiple semantic concepts (polysemantic). In particular, we first manually annotated the 160 test samples for "Dog" for four semantic concepts, for which we knew we had concept vectors in our dictionary, namely "Hot dog" (Concept 2, row 1, column 2 in Fig. 7), "Black dog" (Concept 20, row 10, column 2 in Fig. 7), "Brown/orange dog" (Concept 6, row 3, column 2 in Fig. 7), and "Bull dog" (Concept 15, row 8, column 1 in Fig. 7). For a given semantic concept, we call this set $C_{true}$. Then, for its corresponding concept vector $u_k$ we

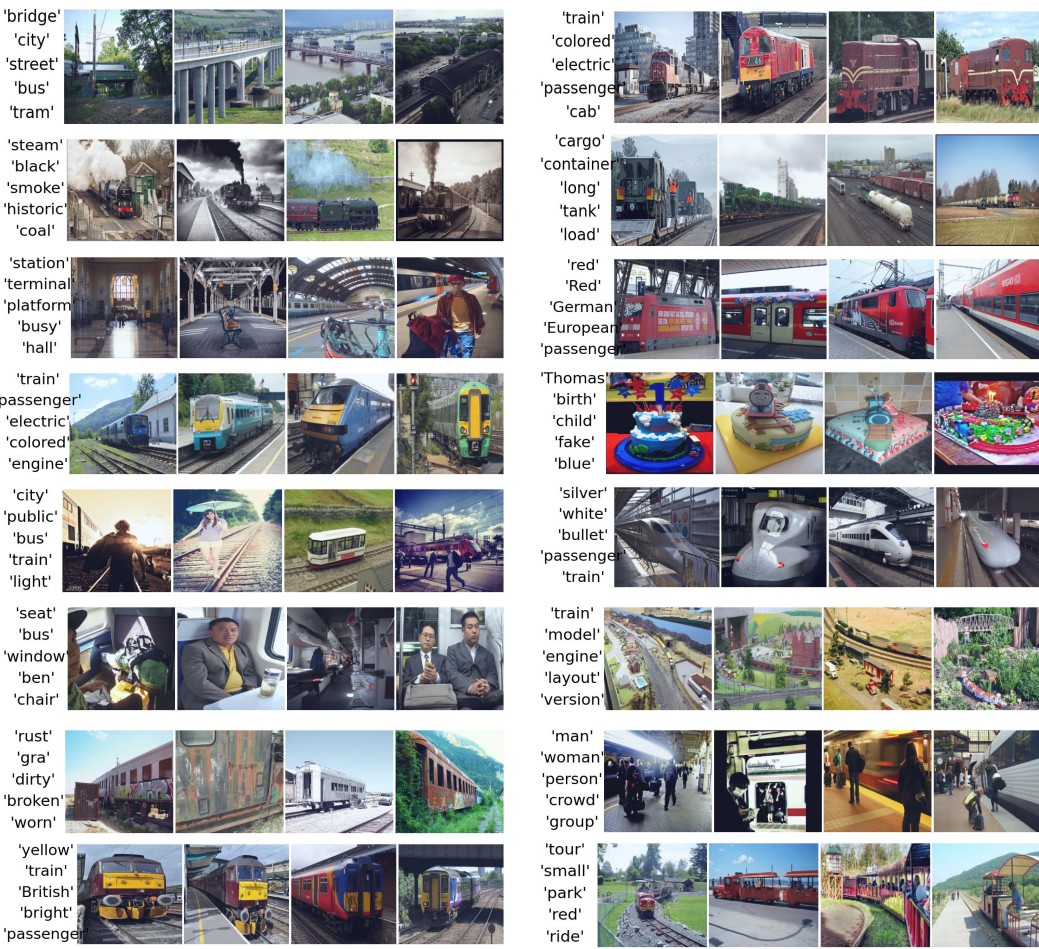

Figure 10: Multimodal grounding for example concepts for 'Train' token (layer 31) on LLaVA.

find the set of test samples for which $u_k$ activates greater than a threshold $\tau$. This threshold was set to half of its maximum activation over test samples. We call this set of samples $C_{top}$. To estimate specificity of the concept vector we compute how many samples in $C_{top}$ lie in the ground-truth set, i.e. $|C_{top}| \cap |C_{true}|/|C_{top}|$.

We found Concept 2 ("Hot dog") to be most monosemantic with 100% specificity. For Concept 20 ("Black dog") too, we found high specificity of 93.3%. For concept 15 ("Bull dog") we observed the lowest specificity of 50%. This concept also activated for test images with toy/stuffed dogs. Interestingly, the multimodal grounding of concept 15 already indicates this superposition with maximum activating samples also containing images of 'toy dogs'. Concept 6 ("Brown/orange dog") is somewhere in between, with 76% specificity. This concept vector also activated sometimes for dark colored dogs, which wasn't apparent from its multimodal grounding.

Prominent or distinct semantic concepts seem to be captured by more specific/monosemantic concept vectors, while more sparsely present concepts seem at risk to be superposed resulting in a more polysemantic concept vector capturing them. It is also worth noting that the multimodal grounding can be a useful tool in some cases to identify polysemanticity in advance.

| Split | Dog | Bus | Train | Cat | Baby | Car | Cake | Bear |
|-------|------|------|------|------|------|------|------|------|
| Train | 3693 | 2382 | 3317 | 3277 | 837 | 1329 | 1733 | 1529 |
| Test | 161 | 91 | 147 | 167 | 44 | 79 | 86 | 55 |

Table 7: Number of samples training/testing samples for each token for DePALM

## C  Further implementation details

### C.1  Dictionary learning details

The details about the number of samples used for training the concept dictionary of each token, and the number of samples for testing is given in Tab. 7. The token representations are of dimension $B = 4096$.

The hyperparameters for the dictionary learning methods are already discussed in the main paper. All the dictionary learning methods (PCA, KMeans, Semi-NMF) are implemented using scikit-learn [37]. For PCA and KMeans we rely on the default optimization strategies. Semi-NMF is implemented through the DictionaryLearning() class, by forcing a positive code. It utilizes the coordinate descent algorithm for optimization during both the learning of $\mathbf{U}^*, \mathbf{V}^*$ and the projection of test representations $v(X)$.

### C.2  CLIPScore/BERTScore evaluation

For a given image $X$ and set of words $\mathbf{T}_k$ associated to concept $u_k$, CLIPScore is calculated between CLIP-image embedding of $X$ and CLIP-text embedding of comma-separated words in $\mathbf{T}_k$. We consider a maximum of 10 most probable words in each $\mathbf{T}_k$, filtering out non-English and stop words. The computation of the metric from embeddings adheres to the standard procedure described in [20]. Our adapted implementation is based on the CLIPScore official repository, which utilizes the ViT-B/32 CLIP model to generate embeddings.

We found that computing BERTScores from comma-separated words and captions is unreliable. Instead, we adopted a method using the LLaMA-3-8B instruct model to construct coherent phrases from a set of grounded words, $\mathbf{T}_k$. Specifically, we provide the LLaMA model with instructions to describe a scene using a designated set of words, for which we also supply potential answers. This instruction is similarly applied to another set of words, but without providing answers. The responses generated by LLaMA are then compared to the captions $y$ using BERTScore. The instruction phrase and an example of the output are detailed in 8. The highest matching score between the generated phrases and the captions of a test sample determines the score assigned to the concept $u_k$. This approach ensures that the evaluation accurately reflects coherent and contextually integrated language use. The metric calculation from embeddings follows the established guidelines outlined in [50]. Our adapted implementation is based on BERTScore official repository, and we use the default Roberta-large model to generate embeddings.

### C.3  Resources

**DePALM experiments compute**   Each experiment to analyze a token with a selected dictionary learning method is conduced on a single RTX5000 (24GB)/ RTX6000 (48GB)/ TITAN-RTX (24GB) GPU. Within dictionary learning, generating visualizations and projecting test data, the majority of time is spent in loading the data/models and extracting the representations. For analysis of a single token with $\approx 3000$ training samples, it takes around 10-15 mins for this complete process. Evaluation for CLIPScore/BERTScore are also conducted using the same resources. Evaluating CLIPScore for 500 (image, grounded-words) pairs takes around 5 mins. The BERTScore evaluation is in contrast more expensive, consuming around 150 mins for 500 pairs.

**LLaVA experiments compute**   Each experiment to extract a concept dictionary for LLaVA was conducted on a single A100 (80GB) GPU. Representation extraction process for LLaVA is more

Table 8: Generating contextually and grammatically coherent phrases Using the LLaMA Model for BERTScore Evaluation

| Instruction to LLaMA | Nature of Response |
|---|---|
| Generate three distinct phrases, each incorporating the words dog, white, people. Ensure each phrase is clear and contextually meaningful. Number each phrase as follows: 1. 2. 3. | 1.A white dog is standing next to people. 2. People are standing next to a white dog. 3. The dog is standing next to people wearing white clothes. |
| Generate three distinct phrases, each incorporating the words List of words . Ensure each phrase is clear and contextually meaningful. Number each phrase as follows: 1. 2. 3. | LLaMA autonomously creates a relevant description, demonstrating comprehension and creative integration of the new words. |
| Generate three distinct phrases, each incorporating the words brown, black, large, fluffy, cat . Ensure each phrase is clear and contextually meaningful. Number each phrase as follows: 1. 2. 3. | 1. A large, fluffy black cat is sleeping on the brown couch. 2. The brown cat is curled up next to a large, fluffy black cat. 3. The large, fluffy cat's brown fur stands out against the black background. |

expensive compared to DePALM consuming around 90 mins for $\approx 3000$ samples. The remaining aspects of dictionary learning, multimodal grounding, representation projection etc. remains relatively cheap. The CLIPScore/BERTScore evaluations are completed with same resources as before.

**Licenses of assets**    The part of the code for representation extraction from LMM is implemented using PyTorch [36]. For our analyses, we also employ the OPT-6.7B model [49] from Meta AI, released under the MIT license, and the CLIP model [39] from OpenAI, available under a custom usage license. Additionally, the COCO dataset [27] used for validation is accessible under the Creative Commons Attribution 4.0 License. We also use CLIPScore [20] and BERTScore [50] for evaluating our method, both publicly released under MIT license. All utilized resources comply with their respective licenses, ensuring ethical usage and reproducibility of our findings.

### C.4    Choice for number of concepts $K$

Our choice of using $K = 20$ concepts for all tokens was driven by the behaviour of reconstruction error of Semi-NMF on the training samples with different values of $K$, i.e. $||\mathbf{Z} - \mathbf{UV}||_2^2$. We validate this behaviour on the four target tokens from main text in Fig. 11. We generally found $K = 20$ as the minimal number of concepts where the reconstruction error drops by at least 50% from $K = 0$.

We also conducted an ablation study to test how our method behaves with different values of number of concepts $K$. Fig. 12 presents the variation of test CLIPScore and Overlap score for $K \in \{10, 20, 30, 50\}$ for two target tokens, 'Dog' and 'Cat'. Our method can learn meaningful concepts for different values of $K$, evident by the consistently high CLIPScore. The Overlap score on the other hand tends to drop more consistently as the number of concepts increase but behaves stably for different choices. Nonetheless they indicate that our method can accommodate dictionaries of larger sizes without compromising the quality of learnt concepts, provided $K << M$ (number of decomposed samples) and at the expense of greater user overhead.

## D    Evaluation and extension to more tokens

We provide test data mean CLIPScore and BERTScore for top-1 activating concept for all baselines and more tokens: Baby, Car, Cake, and Bear in Tab. 9 (results in the main paper are reported for tokens Dog, Bus, Train, Cat in Tab. 1). Additionally, we also report the macro average over a set of 30 additional COCO-nouns apart from the 8 tokens with individually reported results, denoted

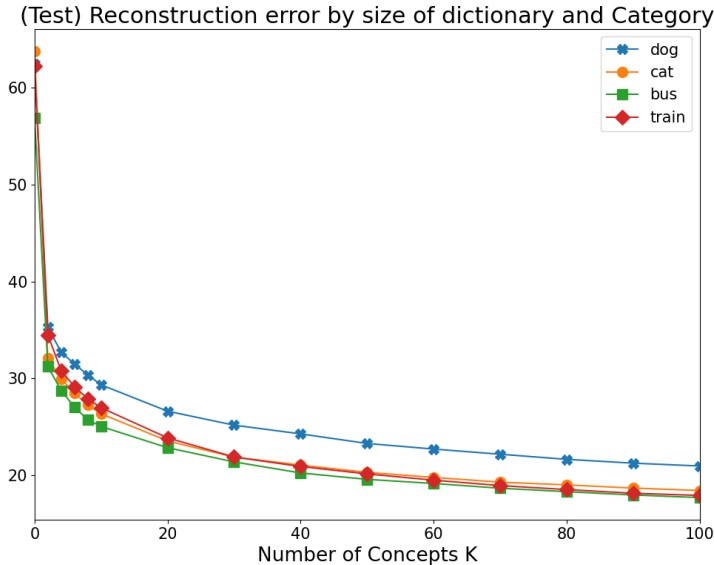

Figure 11: Variation of reconstruction error with number of concepts $K$ for decompositions on different target tokens.

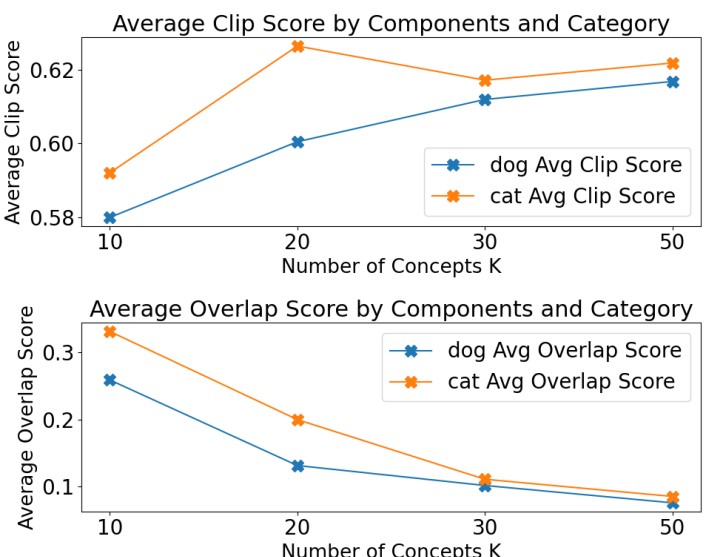

Figure 12: Test CLIPScore and Overlap score ablation with number of concepts $K$. CLIPScore remains consistently high and drops slightly only for very small $K$. Overlap score generally improves with higher $K$.

as 'Extra-30'. These nouns are single-token words with at least 40 predicted test samples. We put the filter of single-token words to keep consistency with the presented framework. Extension to multi-token words is straightforward but discussed separately in D.1. The lower bound criterion on test samples is to ensure average test CLIPScore is reliable for each target token. We only report CLIPScore for 'Extra-30' tokens as BERTScore evaluation was more expensive to conduct on large number of dictionaries.

We observe that we consistently obtain higher scores across for Semi-NMF and K-Means. We also report the overlap score between grounded words in Tab. 10 to illustrate the superiority of our method over the simple baseline. As previously noted, we observe a high overlap between grounded

words with KMeans and Simple baselines compared to Semi-NMF/PCA. A low overlap should be encouraged, as it indicates the discovery of diverse and disentangled concepts in the dictionary. Among all the methods, Semi-NMF provides the most balanced concept dictionaries which are both meaningful (high CLIPScore/BERTScore) and diverse (low overlap).

| Token | Metric | Rnd-Words | Noise-Imgs | Simple | PCA (Ours) | KMeans (Ours) | Semi-NMF (Ours) | GT-captions |
|---|---|---|---|---|---|---|---|---|
| Dog | CS top-1 ($\uparrow$) | $0.519 \pm 0.05$ | $0.425 \pm 0.06$ | $0.546 \pm 0.08$ | $0.559 \pm 0.06$ | $\underline{0.599} \pm 0.07$ | $\mathbf{0.610} \pm 0.09$ | $0.783 \pm 0.06$ |
| | BS top-1 ($\uparrow$) | $0.201 \pm 0.04$ | $0.306 \pm 0.05$ | $0.346 \pm 0.08$ | $0.353 \pm 0.10$ | $\underline{0.398} \pm 0.06$ | $\mathbf{0.405} \pm 0.07$ | $0.511 \pm 0.11$ |
| Bus | CS top-1 ($\uparrow$) | $0.507 \pm 0.05$ | $0.425 \pm 0.08$ | $\mathbf{0.667} \pm 0.06$ | $0.509 \pm 0.05$ | $\underline{0.645} \pm 0.08$ | $0.634 \pm 0.08$ | $0.736 \pm 0.05$ |
| | BS top-1 ($\uparrow$) | $0.200 \pm 0.05$ | $0.303 \pm 0.06$ | $0.390 \pm 0.05$ | $0.380 \pm 0.13$ | $\underline{0.401} \pm 0.06$ | $\mathbf{0.404} \pm 0.07$ | $0.466 \pm 0.11$ |
| Train | CS top-1 ($\uparrow$) | $0.496 \pm 0.05$ | $0.410 \pm 0.07$ | $0.642 \pm 0.06$ | $0.554 \pm 0.08$ | $\mathbf{0.657} \pm 0.06$ | $\underline{0.646} \pm 0.07$ | $0.727 \pm 0.05$ |
| | BS top-1 ($\uparrow$) | $0.210 \pm 0.06$ | $0.253 \pm 0.06$ | $\mathbf{0.392} \pm 0.07$ | $0.334 \pm 0.09$ | $0.375 \pm 0.07$ | $\underline{0.378} \pm 0.07$ | $0.436 \pm 0.08$ |
| Cat | CS top-1 ($\uparrow$) | $0.539 \pm 0.04$ | $0.461 \pm 0.04$ | $0.589 \pm 0.07$ | $0.541 \pm 0.08$ | $\underline{0.608} \pm 0.08$ | $\mathbf{0.627} \pm 0.06$ | $0.798 \pm 0.05$ |
| | BS top-1 ($\uparrow$) | $0.207 \pm 0.07$ | $0.307 \pm 0.03$ | $\underline{0.425} \pm 0.10$ | $0.398 \pm 0.13$ | $0.398 \pm 0.08$ | $\mathbf{0.437} \pm 0.08$ | $0.544 \pm 0.1$ |
| Baby | CS top-1 ($\uparrow$) | $0.532 \pm 0.04$ | $0.471 \pm 0.05$ | $\underline{0.631} \pm 0.06$ | $0.575 \pm 0.07$ | $\mathbf{0.636} \pm 0.05$ | $0.621 \pm 0.06$ | $0.811 \pm 0.05$ |
| | BS top-1 ($\uparrow$) | $0.192 \pm 0.03$ | $0.379 \pm 0.05$ | $\mathbf{0.471} \pm 0.06$ | $0.338 \pm 0.06$ | $0.405 \pm 0.07$ | $\underline{0.426} \pm 0.07$ | $0.530 \pm 0.14$ |
| Car | CS top-1 ($\uparrow$) | $0.518 \pm 0.05$ | $0.461 \pm 0.08$ | $\underline{0.605} \pm 0.04$ | $0.547 \pm 0.08$ | $0.602 \pm 0.05$ | $\mathbf{0.614} \pm 0.05$ | $0.766 \pm 0.06$ |
| | BS top-1 ($\uparrow$) | $0.192 \pm 0.03$ | $0.336 \pm 0.05$ | $\mathbf{0.448} \pm 0.07$ | $0.370 \pm 0.10$ | $\underline{0.435} \pm 0.08$ | $0.379 \pm 0.08$ | $0.485 \pm 0.14$ |
| Cake | CS top-1 ($\uparrow$) | $0.488 \pm 0.05$ | $0.473 \pm 0.08$ | $\underline{0.631} \pm 0.05$ | $0.540 \pm 0.07$ | $\mathbf{0.657} \pm 0.06$ | $0.628 \pm 0.08$ | $0.772 \pm 0.05$ |
| | BS top-1 ($\uparrow$) | $0.186 \pm 0.04$ | $\underline{0.366} \pm 0.07$ | $\mathbf{0.375} \pm 0.10$ | $0.243 \pm 0.08$ | $0.334 \pm 0.07$ | $0.334 \pm 0.08$ | $0.414 \pm 0.13$ |
| Bear | CS top-1 ($\uparrow$) | $0.526 \pm 0.04$ | $0.526 \pm 0.06$ | $0.651 \pm 0.04$ | $0.564 \pm 0.06$ | $\mathbf{0.680} \pm 0.05$ | $\underline{0.660} \pm 0.07$ | $0.798 \pm 0.06$ |
| | BS top-1 ($\uparrow$) | $0.255 \pm 0.10$ | $0.396 \pm 0.08$ | $0.434 \pm 0.05$ | $0.420 \pm 0.10$ | $\underline{0.474} \pm 0.08$ | $\mathbf{0.494} \pm 0.10$ | $0.541 \pm 0.10$ |
| Extra-30 | CS top-1 ($\uparrow$) | $0.516 \pm 0.04$ | $0.521 \pm 0.03$ | $0.626 \pm 0.06$ | $0.547 \pm 0.06$ | $\mathbf{0.637} \pm 0.06$ | $\underline{0.631} \pm 0.06$ | $0.763 \pm 0.05$ |

Table 9: Test data mean CLIPScore and BERTScore for top-1 activating concept CLIPScore denoted as CS, BERTScore denoted as BS for all concept extraction baselines considered. Analysis for layer $L = 31$. Best score indicated in **bold** and second best is underlined. 'Extra-30' denotes a set of 30 additional single-token COCO nouns (apart from previous 8) with at least 40 predicted test samples by $f_{LM}$. For 'Extra-30' tokens we report the macro average and standard deviation of mean test data CLIPScore, taken over the set of 30 tokens.

| Token | Simple | PCA | KMeans | Semi-NMF |
|---|---|---|---|---|
| Baby | 0.645 | **0.006** | 0.502 | 0.187 |
| Car | 0.246 | **0.001** | 0.322 | 0.097 |
| Cake | 0.415 | **0.005** | 0.398 | 0.147 |
| Bear | 0.556 | **0.002** | 0.360 | 0.203 |
| Extra-30 | 0.443 | **0.050** | 0.452 | 0.156 |

Table 10: Overlap/entanglement between grounded words of learnt concepts for different dictionary learning methods. Results are for additional tokens. Lower is better. Best score indicated in **bold** and second best is underlined. For 'Extra-30' tokens (additional 30 single-token COCO nouns) we report the macro average of Overlap score, taken over the set.

**Statistical significance** The statistical significance of Semi-NMF w.r.t all other baselines and variants, for CLIPScore/BERTScore evaluation to understand representations of test samples is given in Tab/ 11 (for all tokens separately). We report the $p$-values for an independent two sided T-test with null hypothesis that mean performance is the same between Semi-NMF and the respective system. The results for Semi-NMF are almost always significant compared to Rnd-Words, Noise-Imgs, PCA. However for these metrics, Simple baseline, K-Means and Semi-NMF all perform competitively and

better than other systems. Within these three systems the significance depends on the target token, but are often not significant in many cases.

| Token | Metric | Rnd-Words | Noise-Imgs | Simple | PCA | KMeans | GT-captions |
|-------|--------|-----------|------------|--------|-----|--------|-------------|
| Dog | CS top-1 | **< 0.001** | **< 0.001** | **< 0.001** | **< 0.001** | > 0.1 | **< 0.001** |
| | BS top-1 | **< 0.001** | **< 0.001** | **< 0.001** | **< 0.001** | > 0.1 | **< 0.001** |
| Bus | CS top-1 | **< 0.001** | **< 0.001** | **0.002** | **< 0.001** | > 0.1 | **< 0.001** |
| | BS top-1 | **< 0.001** | **< 0.001** | > 0.1 | > 0.1 | > 0.1 | **< 0.001** |
| Train | CS top-1 | **< 0.001** | **< 0.001** | > 0.1 | **< 0.001** | > 0.1 | **< 0.001** |
| | BS top-1 | **< 0.001** | **< 0.001** | 0.08 | **< 0.001** | > 0.1 | **< 0.001** |
| Cat | CS top-1 | **< 0.001** | **< 0.001** | **< 0.001** | **< 0.001** | **0.018** | **< 0.001** |
| | BS top-1 | **< 0.001** | **< 0.001** | > 0.1 | **0.001** | **< 0.001** | **< 0.001** |
| Baby | CS top-1 | **< 0.001** | **< 0.001** | > 0.1 | **0.002** | > 0.1 | **< 0.001** |
| | BS top-1 | **< 0.001** | **< 0.001** | **0.001** | **< 0.001** | > 0.1 | **< 0.001** |
| Car | CS top-1 | **< 0.001** | **< 0.001** | > 0.1 | **< 0.001** | > 0.1 | **< 0.001** |
| | BS top-1 | **< 0.001** | **< 0.001** | **< 0.001** | 0.533 | **< 0.001** | **< 0.001** |
| Cake | CS top-1 | **< 0.001** | **< 0.001** | > 0.1 | **< 0.001** | **0.008** | **< 0.001** |
| | BS top-1 | **< 0.001** | **0.005** | **0.003** | **< 0.001** | >0.1 | **< 0.001** |
| Bear | CS top-1 | **< 0.001** | **< 0.001** | > 0.1 | **< 0.001** | 0.072 | **< 0.001** |
| | BS top-1 | **< 0.001** | **< 0.001** | **0.0001** | **0.0001** | >0.1 | **0.015** |

Table 11: Statistical significance of Semi-NMF w.r.t other baselines for test data CLIP-Score/BERTScore. $p$-values for two sided T-test are reported. Significant values ($p$-value $< 0.05$) are indicated in bold. The values do **not** indicate which system has better mean score but just that if the difference is significant.

## D.1 Extending to multi-token words

The presentation of our approach assumes that our token of interest $t$ is a single token. This poses no issues for words which are represented as single tokens but can raise some questions when we wish to extract concept representations for multi-token words. Our approach however, can be easily adapted to this setting. In particular, we extract representation of last token from first prediction of the multi-token sequence. Note that when filtering the training data for samples where ground-truth caption contains the token of interest, we now search for the complete multi-token sequence. The other aspects of the method remain unchanged. While there can also be other viable strategies, the rationale behind this adaptation is that the last token of our sequence of interest can also combines representations from previous tokens in the sequence. We add below results for such examples in Tab. 12 and 13. We observe behaviour consistent with the previous results with Semi-NMF extracting concept dictionaries with high CLIPScore and low overlap.

| Multi-token word | Rnd-Words | Noise-Imgs | Simple | Semi-NMF | GT-captions |
|------------------|-----------|------------|--------|----------|-------------|
| Traffic light | $0.516 \pm 0.03$ | $0.525 \pm 0.03$ | **0.664** $\pm 0.06$ | 0.634 $\pm 0.05$ | $0.744 \pm 0.04$ |
| Cell phone | $0.542 \pm 0.04$ | $0.547 \pm 0.03$ | 0.598 $\pm 0.04$ | **0.598** $\pm 0.05$ | $0.765 \pm 0.06$ |
| Stop sign | $0.533 \pm 0.03$ | $0.549 \pm 0.03$ | **0.617** $\pm 0.08$ | 0.616 $\pm 0.05$ | $0.775 \pm 0.04$ |

Table 12: Test mean CLIPScore ($\uparrow$) reported for top-1 activating concept for multi-token words. Higher scores are better. Best score in **bold**, second best is underlined.

| Multi-token word | Simple | PCA | K-Means | Semi-NMF |
|---|---|---|---|---|
| Traffic light | 0.704 | **0.050** | 0.579 | 0.174 |
| Cell phone | 0.623 | **0.051** | 0.746 | 0.164 |
| Stop sign | 0.461 | **0.058** | 0.704 | 0.109 |

Table 13: Overlap score ($\downarrow$) reported for top-1 activating concept for multi-token words. Higher scores are better. Best score in **bold**, second best is underlined.

# E    Additional visualizations

## E.1    Concept grounding

The visual/textual grounding for all tokens in Tab. 1 are given in Figs. 13 ('Dog'), 14 ('Cat'), 15 ('Bus'), 16 ('Train'). All the results extract $K = 20$ concepts from layer $L = 31$. Similar to our analysis for token 'Dog' in main paper, for a variety of target tokens our method extracts diverse and multimodally coherent concepts encoding various aspects related to the token.

## E.2    Local interpretations

Here, we qualitatively analyze the local interpretations of various decomposition methods, including PCA, k-means, semi-NMF, and the simple baseline strategy. We select these four as they produce coherent grounding compared to Rnd-Words and Noise-Img baselines. We decompose test sample representations on our learnt dictionary and visualize the top three activating components. Note that in the case of KMeans and Simple baseline, the projection maps a given test representation to a single element of the concept dictionary, the one closest to it. However, for uniformity we show the three most closest concept vectors for both. Figs. 17, 18, 19, 20, 21 are dedicated to interpretations of a single sample each, for all four concept extraction methods.

The inferences drawn about the behaviour of the four baselines from quantitative metrics can also be observed qualitatively. Semi-NMF, K-Means and 'Simple' baseline, are all effective at extracting grounded words can be associated to a given image. However, both K-Means and 'Simple' display similar behaviour in terms of highly overlapping grounded words across concepts. This behaviour likely arises due to both the baselines mapping a given representation to a single concept/cluster. This limits their capacity to capture the full complexity of data distributions. In contrast, Semi-NMF and PCA utilize the full dictionary to decompose a given representation and thus recover significantly more diverse concepts. PCA in particular demonstrates almost no overlap, likely due to concept vectors being orthogonal. However, the grounded words for it tend to be less coherent with the images. As noted previously, Semi-NMF excels as the most effective method, balancing both aspects by extracting meaningful and diverse concepts.

# F    Qualitative analysis for different layers

We provide a qualitative comparison of multimodal grounding for the token 'dog' across different layers in Fig. 22. As observed in Fig. 6 (main paper), the multimodal nature of token representations for two tokens 'Dog' and 'Cat' starts to appear around layers $L = 20$ to $L = 25$. It is interesting to note that the representations of images still tend to be separated well, as evident by the most activating samples of different concepts. However, until the deeper layers the grounded words often do not correspond well to the visual grounding. This behaviour only appears strongly in deeper layers.

# G    Analysis for visual tokens

Our analysis in main paper was limited to decomposing representations of text tokens in various layers of an LLM, $h^p_{(l)}, p > N_V$. This was particularly because these were the predicted tokens of the multimodal model. Nevertheless, the same method can also be used to understand the information

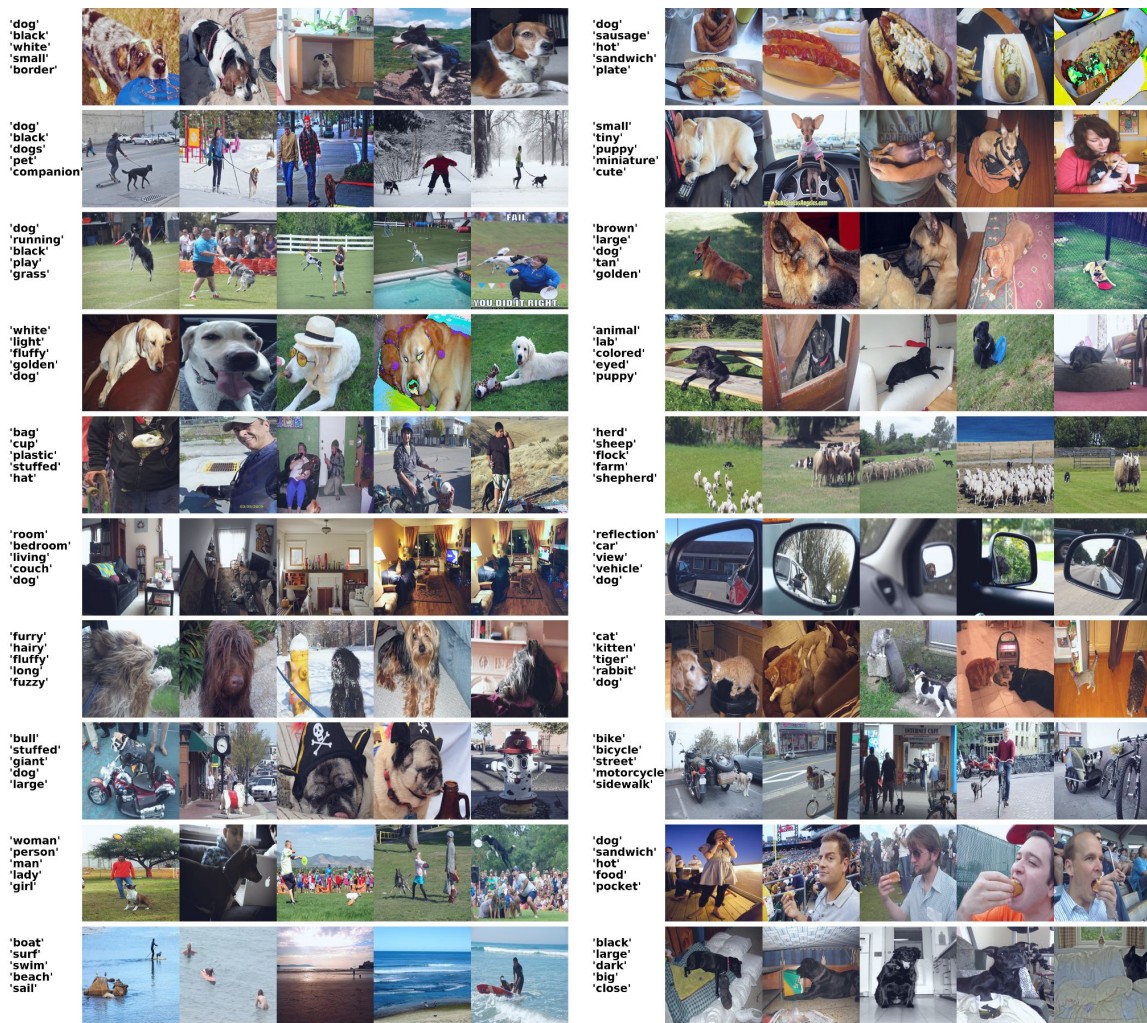

Figure 13: multimodal concept grounding in vision and text for the token 'Dog'. The five most activating samples and the five most probable decoded words for each component $u_k$, $k \in \{1, ..., 20\}$ are shown. The token representations are extracted from L=31 of the LLM section of our LMM.

stored in the visual/perceptual tokens representations as processed in $f_{LM}$, $h_{(l)}^p$, $p \leq N_V$. An interesting aspect worth highlighting is that while the text token representations in $f_{LM}$ can combine information from the visual token representations (via attention function), the reverse is not true. The causal processing structure of $f_{LM}$ prevents the visual token representations to attend to any information in the text token representations. Given a token of interest $t$, for any sample $X \in \mathbf{X}_t$ we now only search for first position $p \in \{1, ..., N_V\}$, s.t. $t = \arg\max \text{Unembed}(h_{(N_L)}^p)$. Only the samples for which such a $p$ exists are considered for decomposition. The rest of the method to learn $\mathbf{U}^*, \mathbf{V}^*$ proceeds exactly as before.

We conduct a small experiment to qualitatively analyze concepts extracted for visual token representations for 'Dog'. We extract $K = 20$ concepts from $L = 31$. The dictionary is learnt with representations from $M = 1752$ samples, less than $M = 3693$ samples for textual tokens. As a brief illustration, 12 out of 20 extracted concepts are shown in Fig. 23. Interestingly, even the visual token representations in deep layers of $f_{LM}$, without ever attending to any text tokens, demonstrate a multimodal semantic structure. It is also worth noting that there are multiple similar concepts that appear for both visual and textual tokens. Concepts 3, 7, 10, 12, 17, 19 are all similar visually and textually to certain concepts discovered for text tokens. This indicates to a possibility that

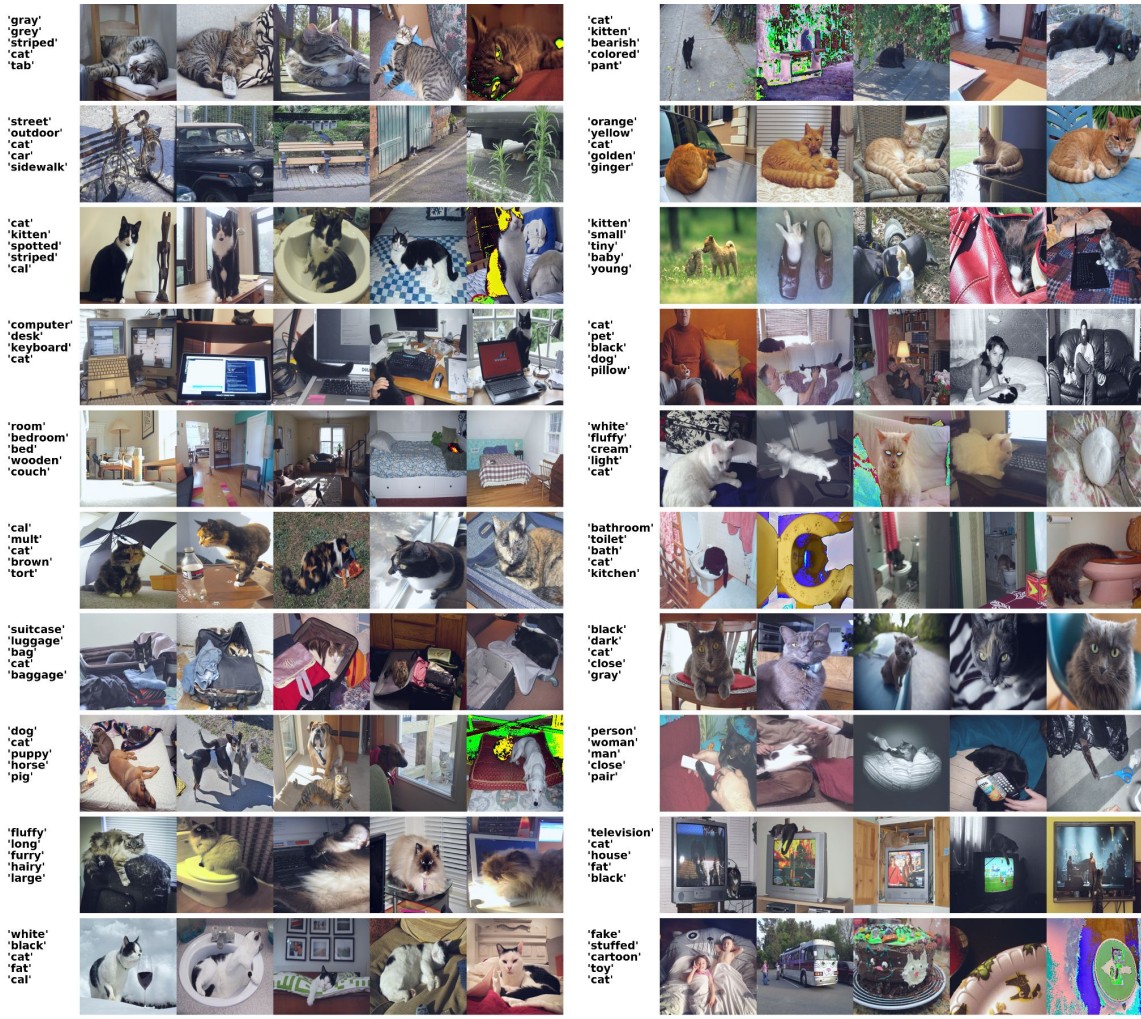

Figure 14: multimodal concept grounding in vision and text for the token 'Cat'. The five most activating samples and the five most probable decoded words for each component $u_k$, $k \in \{1, ..., 20\}$ are shown. The token representations are extracted from L=31 of the LLM section of our LMM.

these concepts are discovered by $f_{LM}$ in processing of the visual tokens and this information gets propagated to predicted text token representations.

## H  Limitations

We list below some limitations of our proposed method:

- The concept dictionaries extracted currently are token-specific. It can be interesting to explore learning concept dictionaries that can encode shared concepts for different tokens.

- The current study is conducted mainly on visual captioning models. While we expect the key ideas to generalize to many other types of large multimodal models, this application of our approach to other types of LMMs remains to be explored/confirmed.

- We select the most simple and straightforward concept grounding techniques. Both visual and textual grounding could potentially be enhanced. The visual grounding can be improved by enhancing localization of concept activation for any MAS or test sample. Text grounding could be enhanced by employing more sophisticated approaches such as tuned lens [6].

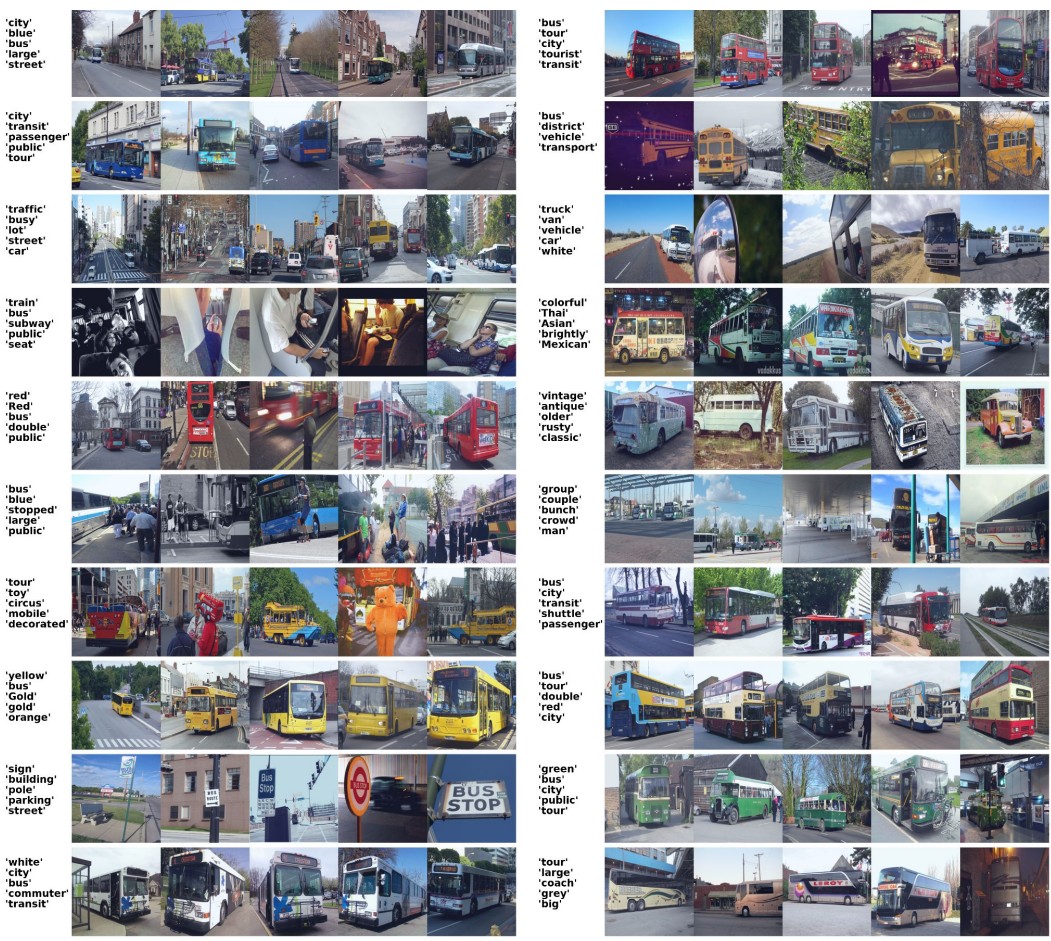

Figure 15: multimodal concept grounding in vision and text for the token 'Bus'. The five most activating samples and the five most probable decoded words for each component $u_k$, $k \in \{1, ..., 20\}$ are shown. The token representations are extracted from L=31 of the LLM section of our LMM.

- While the proposed CLIPScore/BERTScore metrics are useful to validate this aspect, they are not perfect metrics and affected by imperfections and limitations of the underlying models extracting the image/text embeddings. The current research for metrics useful for interpretability remains an interesting open question, even more so in the context of LLMs/LMMs.

# I  Broader societal impact

The popularity of large multimodal models and the applications they are being employed is growing at an extremely rapid pace. The current understanding of these models and their representations is limited, given the limited number of prior methods developed to understand LMMs. Since interpretability is generally regarded as an important trait for machine learning/AI models deployed in real world, we expect our method to have a positive overall impact. This includes its usage for understanding LMMs, as well as encouraging further research in this domain.

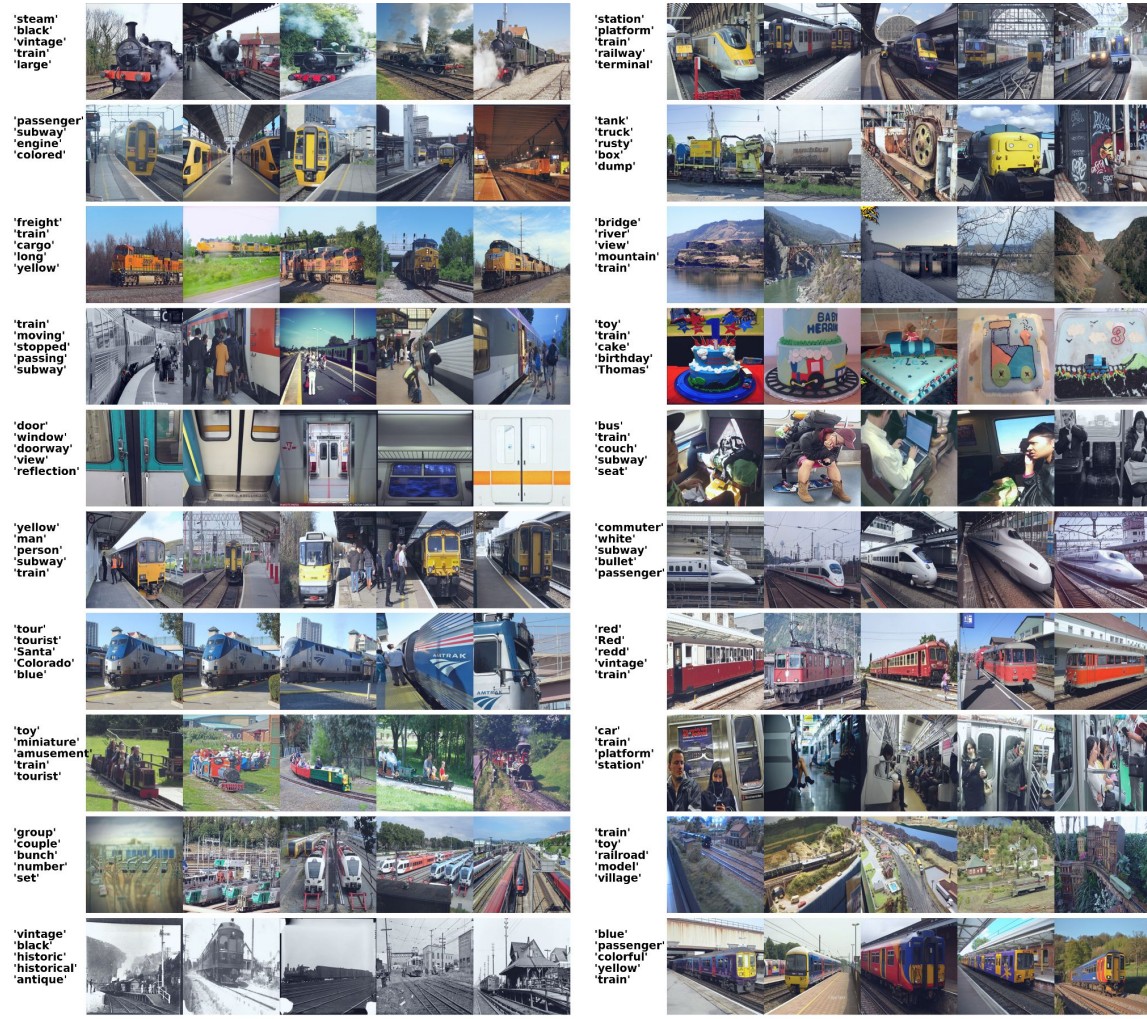

Figure 16: multimodal concept grounding in vision and text for the token 'Train'. The five most activating samples and the five most probable decoded words for each component $u_k$, $k \in \{1, ..., 20\}$ are shown. The token representations are extracted from L=31 of the LLM section of our LMM.


Figure 17: Local interpretations for test sample 9 of token 'Dog' with SemiNMF, KMeans, PCA, and Simple baselines (layer 31). Visual/text grounding for the three highest concept activations (normalized) is shown. SemiNMF baseline provides the most visually and textually consistent results, while other baselines provide components that are not well disentangled (Simple and KMeans baseline), or the text grounding is not closely related to the test image.

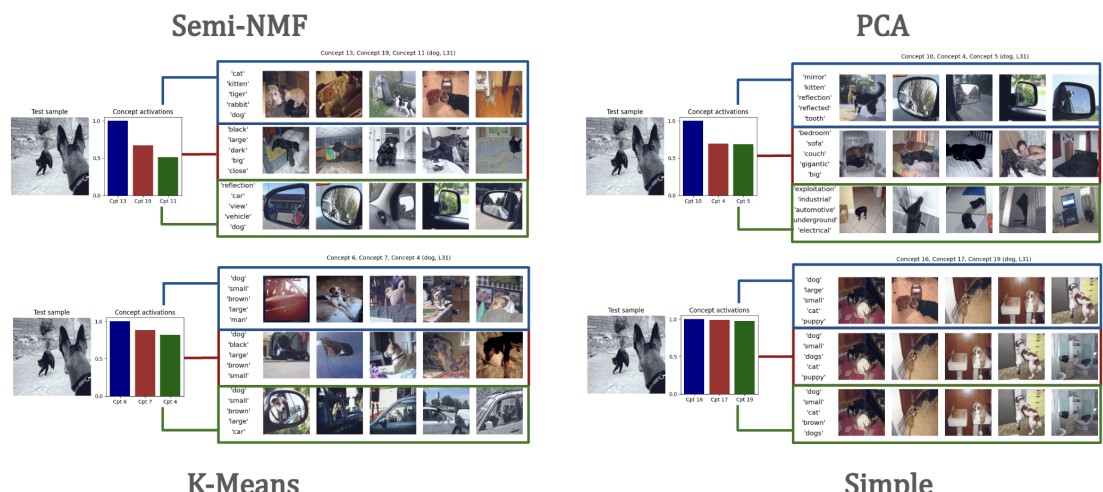

Figure 18: Local interpretations for test sample 37 of token 'Dog' with SemiNMF, KMeans, PCA, and Simple baselines (layer 31). Visual/text grounding for the three highest concept activations (normalized) is shown.

- The claims made should match theoretical and experimental results, and reflect how much the results can be expected to generalize to other settings.
- It is fine to include aspirational goals as motivation as long as it is clear that these goals are not attained by the paper.

2. **Limitations**

Question: Does the paper discuss the limitations of the work performed by the authors?

Answer: [Yes]

Justification: The limitations are discussed separately in Appendix H

Guidelines:

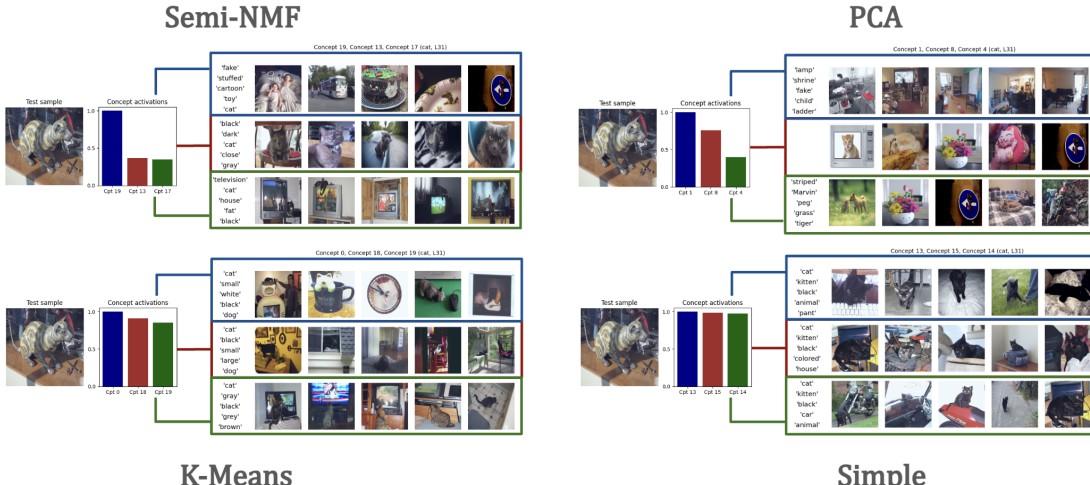

Figure 19: Local interpretations for test sample 43 of token 'Cat' with SemiNMF, KMeans, PCA, and Simple baselines (layer 31). Visual/text grounding for the three highest concept activations (normalized) is shown.

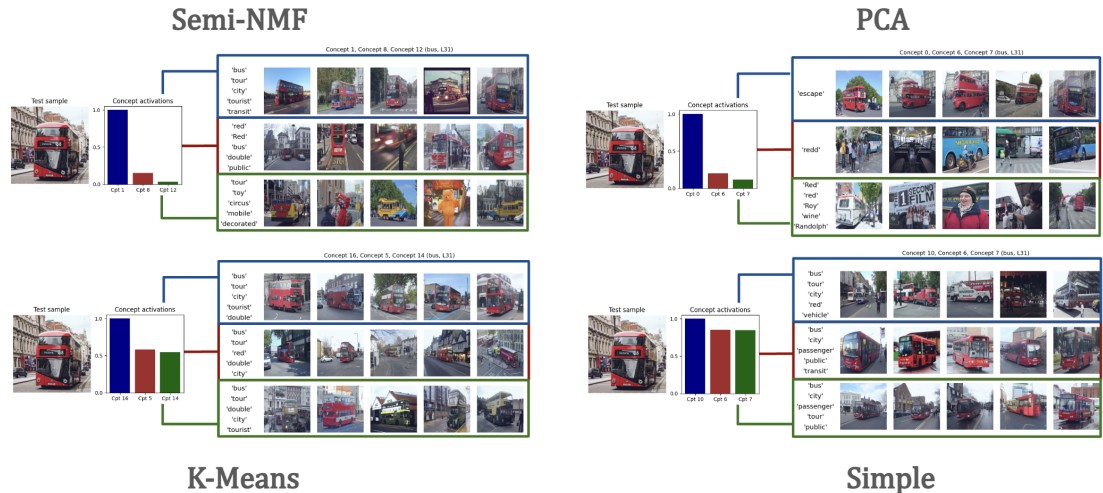

Figure 20: Local interpretations for test sample 6 of token 'Bus' with SemiNMF, KMeans, PCA, and Simple baselines (layer 31). Visual/text grounding for the three highest concept activations (normalized) is shown.

- The answer NA means that the paper has no limitation while the answer No means that the paper has limitations, but those are not discussed in the paper.
- The authors are encouraged to create a separate "Limitations" section in their paper.
- The paper should point out any strong assumptions and how robust the results are to violations of these assumptions (e.g., independence assumptions, noiseless settings, model well-specification, asymptotic approximations only holding locally). The authors should reflect on how these assumptions might be violated in practice and what the implications would be.
- The authors should reflect on the scope of the claims made, e.g., if the approach was only tested on a few datasets or with a few runs. In general, empirical results often depend on implicit assumptions, which should be articulated.
- The authors should reflect on the factors that influence the performance of the approach. For example, a facial recognition algorithm may perform poorly when image resolution

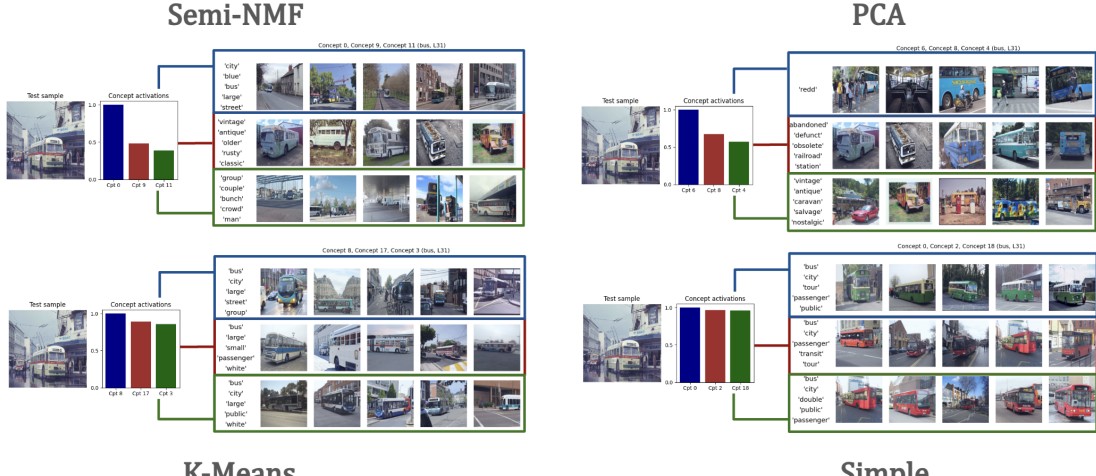

Figure 21: Local interpretations for test sample 12 of token 'Bus' with SemiNMF, KMeans, PCA, and Simple baselines (layer 31). Visual/text grounding for the three highest concept activations (normalized) is shown.

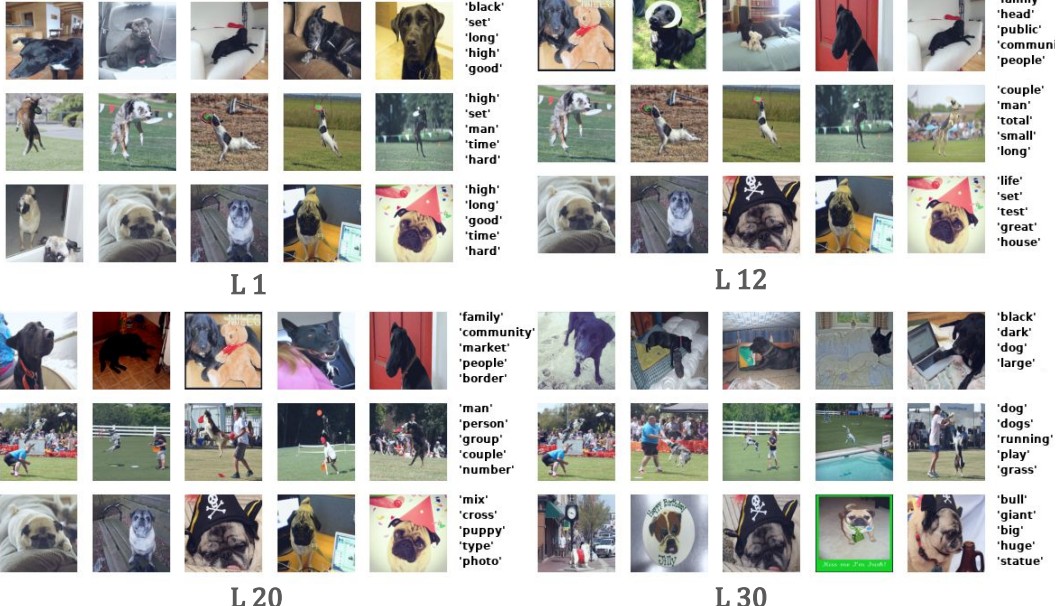

Figure 22: Examples of multimodal grounding across different layers for concepts with similar visual grounding (target token 'Dog'). The grounded words from early layers do not correspond well to the most activating samples of a concept.

is low or images are taken in low lighting. Or a speech-to-text system might not be used reliably to provide closed captions for online lectures because it fails to handle technical jargon.

- The authors should discuss the computational efficiency of the proposed algorithms and how they scale with dataset size.
- If applicable, the authors should discuss possible limitations of their approach to address problems of privacy and fairness.
- While the authors might fear that complete honesty about limitations might be used by reviewers as grounds for rejection, a worse outcome might be that reviewers discover

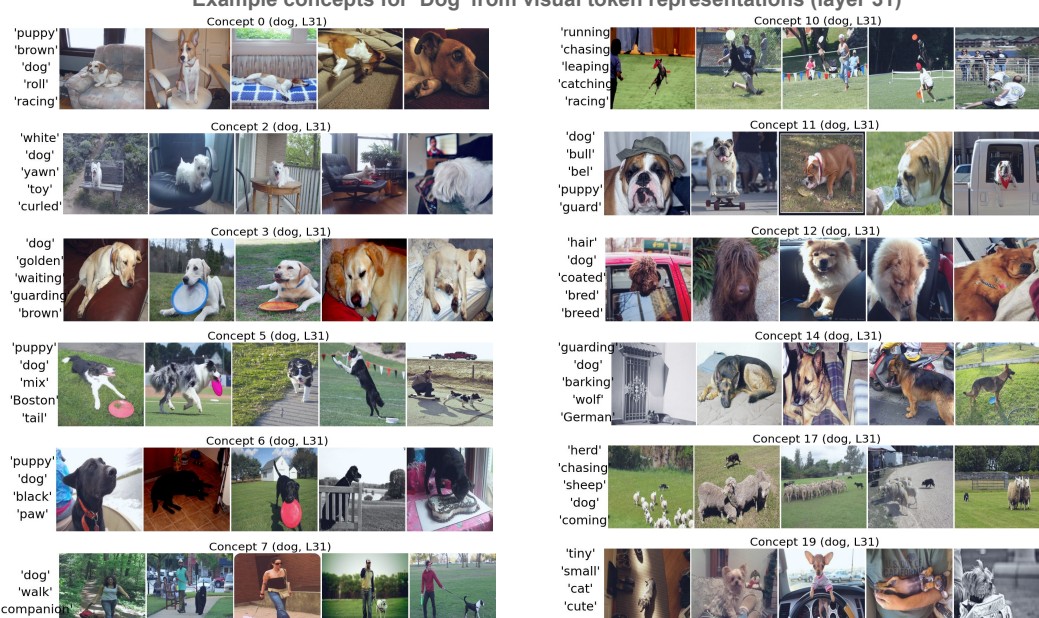

Figure 23: Example concepts extracted for 'Dog' from visual token representations in layer 31.

limitations that aren't acknowledged in the paper. The authors should use their best judgment and recognize that individual actions in favor of transparency play an important role in developing norms that preserve the integrity of the community. Reviewers will be specifically instructed to not penalize honesty concerning limitations.

3. **Theory Assumptions and Proofs**

   Question: For each theoretical result, does the paper provide the full set of assumptions and a complete (and correct) proof?

   Answer: [NA]

   Justification: The paper does not posit any theoretical results. The paper primarily presents empirical evidence demonstrating the effectiveness of the method rather than focusing on theoretical results. our approach highlights the practical applicability and robust performance of the method through a series of experiments.

   Guidelines:

   - The answer NA means that the paper does not include theoretical results.
   - All the theorems, formulas, and proofs in the paper should be numbered and cross-referenced.
   - All assumptions should be clearly stated or referenced in the statement of any theorems.
   - The proofs can either appear in the main paper or the supplemental material, but if they appear in the supplemental material, the authors are encouraged to provide a short proof sketch to provide intuition.
   - Inversely, any informal proof provided in the core of the paper should be complemented by formal proofs provided in appendix or supplemental material.
   - Theorems and Lemmas that the proof relies upon should be properly referenced.

4. **Experimental Result Reproducibility**

   Question: Does the paper fully disclose all the information needed to reproduce the main experimental results of the paper to the extent that it affects the main claims and/or conclusions of the paper (regardless of whether the code and data are provided or not)?

   Answer: [Yes]

Justification: We have disclosed complete information about our pretrained model, extraction of representations, parameters, and implementation to train the dictionary, evaluation setup, and resource usage for the experiments. Details are in Sec. 4 and Appendix C.

Guidelines:

- The answer NA means that the paper does not include experiments.
- If the paper includes experiments, a No answer to this question will not be perceived well by the reviewers: Making the paper reproducible is important, regardless of whether the code and data are provided or not.
- If the contribution is a dataset and/or model, the authors should describe the steps taken to make their results reproducible or verifiable.
- Depending on the contribution, reproducibility can be accomplished in various ways. For example, if the contribution is a novel architecture, describing the architecture fully might suffice, or if the contribution is a specific model and empirical evaluation, it may be necessary to either make it possible for others to replicate the model with the same dataset, or provide access to the model. In general. releasing code and data is often one good way to accomplish this, but reproducibility can also be provided via detailed instructions for how to replicate the results, access to a hosted model (e.g., in the case of a large language model), releasing of a model checkpoint, or other means that are appropriate to the research performed.
- While NeurIPS does not require releasing code, the conference does require all submissions to provide some reasonable avenue for reproducibility, which may depend on the nature of the contribution. For example
  (a) If the contribution is primarily a new algorithm, the paper should make it clear how to reproduce that algorithm.
  (b) If the contribution is primarily a new model architecture, the paper should describe the architecture clearly and fully.
  (c) If the contribution is a new model (e.g., a large language model), then there should either be a way to access this model for reproducing the results or a way to reproduce the model (e.g., with an open-source dataset or instructions for how to construct the dataset).
  (d) We recognize that reproducibility may be tricky in some cases, in which case authors are welcome to describe the particular way they provide for reproducibility. In the case of closed-source models, it may be that access to the model is limited in some way (e.g., to registered users), but it should be possible for other researchers to have some path to reproducing or verifying the results.

5. **Open access to data and code**

   Question: Does the paper provide open access to the data and code, with sufficient instructions to faithfully reproduce the main experimental results, as described in supplemental material?

   Answer: [Yes]

   Justification: Our code is available at `https://github.com/mshukor/xl-vlms`

   Guidelines:

   - The answer NA means that paper does not include experiments requiring code.
   - Please see the NeurIPS code and data submission guidelines (`https://nips.cc/public/guides/CodeSubmissionPolicy`) for more details.
   - While we encourage the release of code and data, we understand that this might not be possible, so "No" is an acceptable answer. Papers cannot be rejected simply for not including code, unless this is central to the contribution (e.g., for a new open-source benchmark).
   - The instructions should contain the exact command and environment needed to run to reproduce the results. See the NeurIPS code and data submission guidelines (`https://nips.cc/public/guides/CodeSubmissionPolicy`) for more details.
   - The authors should provide instructions on data access and preparation, including how to access the raw data, preprocessed data, intermediate data, and generated data, etc.

- The authors should provide scripts to reproduce all experimental results for the new proposed method and baselines. If only a subset of experiments are reproducible, they should state which ones are omitted from the script and why.
- At submission time, to preserve anonymity, the authors should release anonymized versions (if applicable).
- Providing as much information as possible in supplemental material (appended to the paper) is recommended, but including URLs to data and code is permitted.

6. **Experimental Setting/Details**

    Question: Does the paper specify all the training and test details (e.g., data splits, hyper-parameters, how they were chosen, type of optimizer, etc.) necessary to understand the results?

    Answer: [Yes]

    Justification: Major details are in Sec. 4 and all additional details in Appendix C.

    Guidelines:
    - The answer NA means that the paper does not include experiments.
    - The experimental setting should be presented in the core of the paper to a level of detail that is necessary to appreciate the results and make sense of them.
    - The full details can be provided either with the code, in appendix, or as supplemental material.

7. **Experiment Statistical Significance**

    Question: Does the paper report error bars suitably and correctly defined or other appropriate information about the statistical significance of the experiments?

    Answer: [Yes]

    Justification: We show standard deviation of the key metrics in Tab. 1 and Tab. 9 and discuss the statistical significance comparing baselines and our method in Appendix D.

    Guidelines:
    - The answer NA means that the paper does not include experiments.
    - The authors should answer "Yes" if the results are accompanied by error bars, confidence intervals, or statistical significance tests, at least for the experiments that support the main claims of the paper.
    - The factors of variability that the error bars are capturing should be clearly stated (for example, train/test split, initialization, random drawing of some parameter, or overall run with given experimental conditions).
    - The method for calculating the error bars should be explained (closed form formula, call to a library function, bootstrap, etc.)
    - The assumptions made should be given (e.g., Normally distributed errors).
    - It should be clear whether the error bar is the standard deviation or the standard error of the mean.
    - It is OK to report 1-sigma error bars, but one should state it. The authors should preferably report a 2-sigma error bar than state that they have a 96% CI, if the hypothesis of Normality of errors is not verified.
    - For asymmetric distributions, the authors should be careful not to show in tables or figures symmetric error bars that would yield results that are out of range (e.g. negative error rates).
    - If error bars are reported in tables or plots, The authors should explain in the text how they were calculated and reference the corresponding figures or tables in the text.

8. **Experiments Compute Resources**

    Question: For each experiment, does the paper provide sufficient information on the computer resources (type of compute workers, memory, time of execution) needed to reproduce the experiments?

    Answer: [Yes]

    Justification: Details are in Appendix C.

Guidelines:

- The answer NA means that the paper does not include experiments.
- The paper should indicate the type of compute workers CPU or GPU, internal cluster, or cloud provider, including relevant memory and storage.
- The paper should provide the amount of compute required for each of the individual experimental runs as well as estimate the total compute.
- The paper should disclose whether the full research project required more compute than the experiments reported in the paper (e.g., preliminary or failed experiments that didn't make it into the paper).

9. **Code Of Ethics**

Question: Does the research conducted in the paper conform, in every respect, with the NeurIPS Code of Ethics https://neurips.cc/public/EthicsGuidelines?

Answer: [Yes]

Justification: We have reviewed them and adhere to them.

Guidelines:

- The answer NA means that the authors have not reviewed the NeurIPS Code of Ethics.
- If the authors answer No, they should explain the special circumstances that require a deviation from the Code of Ethics.
- The authors should make sure to preserve anonymity (e.g., if there is a special consideration due to laws or regulations in their jurisdiction).

10. **Broader Impacts**

Question: Does the paper discuss both potential positive societal impacts and negative societal impacts of the work performed?

Answer: [Yes]

Justification: The societal impacts are discussed separately in Appendix I.

Guidelines:

- The answer NA means that there is no societal impact of the work performed.
- If the authors answer NA or No, they should explain why their work has no societal impact or why the paper does not address societal impact.
- Examples of negative societal impacts include potential malicious or unintended uses (e.g., disinformation, generating fake profiles, surveillance), fairness considerations (e.g., deployment of technologies that could make decisions that unfairly impact specific groups), privacy considerations, and security considerations.
- The conference expects that many papers will be foundational research and not tied to particular applications, let alone deployments. However, if there is a direct path to any negative applications, the authors should point it out. For example, it is legitimate to point out that an improvement in the quality of generative models could be used to generate deepfakes for disinformation. On the other hand, it is not needed to point out that a generic algorithm for optimizing neural networks could enable people to train models that generate Deepfakes faster.
- The authors should consider possible harms that could arise when the technology is being used as intended and functioning correctly, harms that could arise when the technology is being used as intended but gives incorrect results, and harms following from (intentional or unintentional) misuse of the technology.
- If there are negative societal impacts, the authors could also discuss possible mitigation strategies (e.g., gated release of models, providing defenses in addition to attacks, mechanisms for monitoring misuse, mechanisms to monitor how a system learns from feedback over time, improving the efficiency and accessibility of ML).

11. **Safeguards**

Question: Does the paper describe safeguards that have been put in place for responsible release of data or models that have a high risk for misuse (e.g., pretrained language models, image generators, or scraped datasets)?

Answer: [NA]

Justification: We work with models and data already publicly available.

Guidelines:

- The answer NA means that the paper poses no such risks.
- Released models that have a high risk for misuse or dual-use should be released with necessary safeguards to allow for controlled use of the model, for example by requiring that users adhere to usage guidelines or restrictions to access the model or implementing safety filters.
- Datasets that have been scraped from the Internet could pose safety risks. The authors should describe how they avoided releasing unsafe images.
- We recognize that providing effective safeguards is challenging, and many papers do not require this, but we encourage authors to take this into account and make a best faith effort.

12. **Licenses for existing assets**

    Question: Are the creators or original owners of assets (e.g., code, data, models), used in the paper, properly credited and are the license and terms of use explicitly mentioned and properly respected?

    Answer: [Yes]

    Justification: Details available in Appendix C.

    Guidelines:

    - The answer NA means that the paper does not use existing assets.
    - The authors should cite the original paper that produced the code package or dataset.
    - The authors should state which version of the asset is used and, if possible, include a URL.
    - The name of the license (e.g., CC-BY 4.0) should be included for each asset.
    - For scraped data from a particular source (e.g., website), the copyright and terms of service of that source should be provided.
    - If assets are released, the license, copyright information, and terms of use in the package should be provided. For popular datasets, `paperswithcode.com/datasets` has curated licenses for some datasets. Their licensing guide can help determine the license of a dataset.
    - For existing datasets that are re-packaged, both the original license and the license of the derived asset (if it has changed) should be provided.
    - If this information is not available online, the authors are encouraged to reach out to the asset's creators.

13. **New Assets**

    Question: Are new assets introduced in the paper well documented and is the documentation provided alongside the assets?

    Answer: [Yes]

    Justification: We publicly release our code and documentation is available alongside it.

    Guidelines:

    - The answer NA means that the paper does not release new assets.
    - Researchers should communicate the details of the dataset/code/model as part of their submissions via structured templates. This includes details about training, license, limitations, etc.
    - The paper should discuss whether and how consent was obtained from people whose asset is used.
    - At submission time, remember to anonymize your assets (if applicable). You can either create an anonymized URL or include an anonymized zip file.

14. **Crowdsourcing and Research with Human Subjects**

Question: For crowdsourcing experiments and research with human subjects, does the paper include the full text of instructions given to participants and screenshots, if applicable, as well as details about compensation (if any)?

Answer: [NA]

Justification:

Guidelines:

- The answer NA means that the paper does not involve crowdsourcing nor research with human subjects.
- Including this information in the supplemental material is fine, but if the main contribution of the paper involves human subjects, then as much detail as possible should be included in the main paper.
- According to the NeurIPS Code of Ethics, workers involved in data collection, curation, or other labor should be paid at least the minimum wage in the country of the data collector.

15. **Institutional Review Board (IRB) Approvals or Equivalent for Research with Human Subjects**

Question: Does the paper describe potential risks incurred by study participants, whether such risks were disclosed to the subjects, and whether Institutional Review Board (IRB) approvals (or an equivalent approval/review based on the requirements of your country or institution) were obtained?

Answer: [NA]

Justification:

Guidelines:

- The answer NA means that the paper does not involve crowdsourcing nor research with human subjects.
- Depending on the country in which research is conducted, IRB approval (or equivalent) may be required for any human subjects research. If you obtained IRB approval, you should clearly state this in the paper.
- We recognize that the procedures for this may vary significantly between institutions and locations, and we expect authors to adhere to the NeurIPS Code of Ethics and the guidelines for their institution.
- For initial submissions, do not include any information that would break anonymity (if applicable), such as the institution conducting the review.

