# OpenReview forum: "A Concept-Based Explainability Framework for Large Multimodal Models"
_NeurIPS.cc/2024/Conference — NeurIPS 2024 poster_

### Official Review · Reviewer_17ab · 2024-07-09

**Soundness:** 3
**Presentation:** 4
**Contribution:** 3
**Rating:** 7
**Confidence:** 3

**Summary:**

This paper introduces an explainability approach for interpreting the internal representations of large multimodal models (LMMs). The authors train an image captioning model consisting of a pretrained image encoder and language model and a connector model. To extract interpretable representations, the authors use dictionary learning, decomposing representations into lower dimensional $U$ and $V$ matrices using semi non-negative matrix factorization as the optimisation objective. To interpret concepts in $U$ in the textual domain, the authors use the language model unembedding layer to extract the highest probability tokens associated with a given concept vector $u$. Likewise, to visually interpret a concept vector $u$, the authors find the set of images that maximally activate $u$. The authors provide quantitative evidence demonstrating that their method generates concepts which are well aligned with both the input image and the ground-truth captions. Additionally, they demonstrate their approach qualitatively produces well defined concepts with limited overlap between the tokens represented by different concepts.

**Strengths:**

1. This paper introduces a promising novel approach to performing mechanistic interpretability on multimodal models. Given the increasing ubiquity of multimodal models there exists a clear need for interpretability approaches for this style of model. This work presents a viable dictionary-learning based approach I look forward to seeing other researchers building upon in the future.
2. The evaluation of the framework is comprehensive including both qualitative and quantitative results, both of which are essential for any interpretability method. Additionally, I was impressed to see the authors consider multiple approaches (e.g. PCA/KMeans) and sensible baseline models when evaluating their results, providing more confidence that the final approach taken was an appropriate method.
3. The paper is very clearly articulated, with the rationale well-defined, the approach clearly outlined and the results succinctly and clearly summarised.

**Weaknesses:**

1. The authors provide good evidence that the concepts extracted by their method show minimal overlap with other concepts, however they do not address the alternative possibility, that their extracted concepts might represent more than one distinct concept. This phenomenon of feature “superposition” has been well documented in other model interpretability work (see [1][2][3]) and so it seems plausible that it may arise in the approach taken here. This seems especially likely given that the dimensionality of their concept dictionary is lower than the dimensionality of the internal representations. Though I do not think this should detract from the otherwise excellent contributions presented in this paper, I do think this at least warrants a brief discussion, and perhaps more qualitative analysis of the extracted concepts, to assess whether any evidence of feature superposition is observed.
2. Occasionally the authors make claims that go beyond the scope of this work. For example, in the abstract they state, “we present a novel framework for the interpretation of LMMs”. However, this claim seems too strong given that this approach is only really valid for image captioning models rather than large multimodal models more generally. Additionally, there are a few comments such as “we find the generalization of LLMs to multimodal inputs is an interesting phenomenon to understand” in 3.1 (under “Training”) and “the multimodal structure of internal token representations starts to appear at [later layers]” in 4.2 (under “Layer Ablation”). However these claims do not seem valid as the language model layers are frozen during training. It seems more appropriate to say that these extracted concepts represent language model concepts, and the connector model learns to transform the image representations such that they align with these language model concepts.

[1] Elhage, N, et al. "Toy Models of Superposition." arXiv:2209.10652 (2022)

[2] Arora, S et al. "Linear Algebraic Structure of Word Senses, with Applications to Polysemy." arXiv:1601.03764 (2016)

[3] Elhage, N, et al. "Softmax Linear Units" Transformer Circuits Thread (2022)

**Questions:**

1. Why did the authors choose to use Opt-6.7B as the language model rather than more powerful similar sized models such as Llama-7B?
2. Why do the authors take the absolute activations of $V$ in (5)? Are these activations not guaranteed to be non-negative by the optimisation objective?
3. Did the authors consider trialling gradient-based feature visualisation approaches [1] in addition to taking images with the highest activations? This could be an alternative approach to build additional confidence that the extracted concepts do represent what qualitative analysis of the highest activating samples appears to suggest. I don’t expect the authors to perform analysis of this kind in this manuscript however it could be worth commenting on this as a future avenue of research, or alternatively raising any valid criticisms of this approach?
4. There is a typo in figure 1 (“Captionin”)

[1] Olah, C, et al. "Feature Visualization", Distill, 2017.

**Limitations:**

The work presented here is only for a single, relatively small model, trained with a specific objective (image captioning). As such it is not clear that this approach will necessarily generalise to other multimodal models. The authors should touch on this limitation in the discussion or limitations section in the appendix.

---

> ### Author Rebuttal · Authors · 2024-08-06
>
> Thank you for the positive and insightful comments. Please find our response pointwise below:
>
> **Qualitative analysis for feature superposition/specificity of concept vectors:** Thanks for the interesting suggestion. We conducted a preliminary qualitative study on some concept vectors in the dictionary learnt for token "Dog" to analyze if these concept vectors tend to activate strongly for a specific semantic concept (monosemantic) or multiple semantic concepts (polysemantic). In particular, we first manually annotated the 160 test samples for "Dog" for four semantic concepts, for which we knew we had concept vectors in our dictionary, namely "Hot dog" (Concept 2, row 1, column 2 in Fig. 7), "Black dog" (Concept 20, row 10, column 2 in Fig. 7), "Brown/orange dog" (Concept 6, row 3, column 2 in Fig. 7), and "Bull dog" (Concept 15, row 8, column 1 in Fig. 7). For a given semantic concept, we call this set $C_{true}$. Then, for its corresponding concept vector $u_k$ we find the set of test samples for which it activates greater than a threshold $\tau$. This threshold was set to half of its maximum activation over test samples. We call this set of samples $C_{top}$. To estimate specificity of the concept vector we compute how many samples in $C_{top}$ lie in the ground-truth set, i.e. $|C_{top} \cap C_{true}|/|C_{top}|$. We find Concept 2 ("Hot dog") to be most monosemantic with 100\% specificity. For Concept 20 ("Black dog") too, we found high specificity of 93.3\%. For concept 15 ("Bull dog") we observed the lowest specificity of 50\%. This concept also activated for test images with toy/stuffed dogs. Interestingly, the multimodal grounding of concept 15 already indicates this superposition with maximum activating samples also containing images of 'toy dogs'. Concept 6 ("Brown/orange dog") is somewhere in between, with 76\% specificity. This concept vector also activated sometimes for dark colored dogs, which wasn't apparent from its multimodal grounding.
>
> In summary, your expectation of superposition/polysemanticity is fair. Prominent or distinct semantic concepts seem to be captured by more specific/monosemantic concept vectors, while more sparsely present concepts seem at risk to be superposed resulting in a more polysemantic concept vector capturing them. We can add this discussion in appendix.
>
>
> **Qualifying certain claims:** We accept your concerns about the three claims/statements. We will modify them to qualify their extent and scope.
>
>
> **Powerful language model + Generalization to other LMMs:** We are pleased to share that we conducted new experiments LLaVA-v1.5. It uses Vicuna-7B (closely related to LLaMA) as the language model. We are able to extract meaningful multimodal concepts and achieve quantitatively consistent results as for DePALM. The global response presents more details regarding the same.
>
>
> **Absolute activations in Eq. (5):** We wrote Eq. (5) this way so that it remains applicable even for methods without the constraint for non-negativity of activations (eg. PCA). For Semi-NMF it indeed does not make a difference.
>
> **Trialling gradient-based feature visualisation:** This is a fair suggestion and a useful future direction to build upon. For now, we have explored generating saliency maps with LLaVA (not via gradients) for concept activations by computing the inner product of concept vector $u_k$ with all visual token representations from corresponding layer $L$, i.e. $u_k^T [h^1_{(L)}, ..., h^{N_V}_{(L)}]$. This vector of size $N_V =576$ is reshaped to 24 $\times$ 24 and resized to original input. This is feasible for LLaVA as all visual token representations fed to the language model preserve their patch based identity. Sample visualizations are available in 1-page PDF (Fig. 2) in global response.
>
> **Typo in Fig. 1:** Thanks for pointing it out. We'll correct it.

---

> > ### Comment · Reviewer_17ab · 2024-08-09
> >
> > Thank-you for addressing all my comments, the additional analyses are very interesting. I’m happy with all the responses to my queries and look forward to the final version of the manuscript. I have one additional comment I’ve made below:
> >
> > **Qualitative analysis for feature superposition/specificity of concept vectors**
> >
> > The polysemanticity analysis is very interesting. Thank-you for taking the time to conduct this analysis, I think it would be a useful discussion point in the appendix. The analysis you have presented considers the specificity of concepts related to a given token (“dog” in this instance). I would be interested in understanding the extent to which the concepts for one token are polysemantic for other tokens e.g. is the “Hot dog” concept active for tokens other than “dog”. I don’t expect you to conduct any additional analyses or to edit the manuscript but just wanted to raise this point as a potential future avenue of research.

---

> > > ### Author Response · Authors · 2024-08-11
> > >
> > > Thanks for the rebuttal acknowledgement! Happy to address all your questions! For future development, we'll certainly keep your point about polysemanticity of concept vectors to other tokens under consideration .

---

### Official Review · Reviewer_afLg · 2024-07-12

**Soundness:** 3
**Presentation:** 3
**Contribution:** 3
**Rating:** 6
**Confidence:** 3

**Summary:**

This paper proposes a new approach to understand multimodal concepts learned in LLMs with visual prefixes. To do so, the authors propose a dictionary learning-based approach that decomposes the representation of a word token in the product of two low-rank matrices via Semi-NMF, one representing the concepts and the other representing the activations for a given token. The authors evaluate the representation of the DePALM model on 8 common objects (e.g., dog, bus, cat) showing qualitatively good results, as well as better auto-eval metrics than with related baselines constructed by the authors.

**Strengths:**

Originality: The paper proposes a novel method to interpret concepts in a multimodal LLM by grounding them in both text and visual spaces. For a given concept, it creates a representation matrix from image–caption pairs, which is then decomposed into two low-rank matrices using Semi-NMF.

Quality: The authors show that extracted concepts for a given token can be interpreted both visually and textually through qualitative examples. Quantitatively, the method outperforms other related baselines introduced by the authors.

Clarity: The paper is relatively well written, although some parts (eg, Section 3.4, 4.0 and 4.1) feel very dense and require more time to be processed.

Significance: The paper provides interesting insights in the representation of a given word token processed by a frozen LLM that is augmented with image understanding via visual prefix learning.

**Weaknesses:**

1. My main concern is in studying the representation of only eight tokens. It would be interesting to study more words and see if there are any interesting takeaways. These could include the 80 COCO classes, and additional rarer tokens.
2. The use of Semi-NMF in L165-167 is unclear. Later on, we find out that it works better than other decompositions, but I was left wondering how the authors came up with Semi-NMF in Section 3.4.
3. It would have been useful to compare the proposed approach with some current interpretability methods that could apply here (or at least discuss why they might not be used here).

**Questions:**

1. Are there any blockers to perform a larger-scale (wrt tokens) analysis?
2. Interpretability methods like ROME use the last token of a given word – why do you use the first token?
3. Does the method actually work for multi-token words (i.e., words that are split into multiple tokens by the tokenizer)? If so, are any of your 8 words in that category?
4. Please double-check your references, some of them appear more than once.

**Limitations:**

Yes.

---

> ### Author Rebuttal · Authors · 2024-08-06
>
> We thank the reviewer for the interesting feedback and positive comments:
>
> **Computational scaling for large number of tokens:** Our experiments use single GPU. The BERTScore evaluation does not scale well. It can take upto 3-4 hours to evaluate all baselines on some target tokens (COCO). The representation extraction process scales fine with DePALM (upto 15 minutes for some tokens), but poorly with models like LLaVA (upto 3-5 hours for some tokens). The core of the method itself, dictionary learning and multimodal grounding, is computationally cheap.
>
> **Experiments with more tokens:**
> We didn't experiment earlier with more because of expensive BERTScore evaluation. However, we have now conducted experiments with 30 additional COCO classes (apart from 8 in the paper) for CLIPScore and Overlap , with identical hyperparameters. We considered these COCO classes based on the criteria that these are single token words with at least 40 predicted test samples by $f_{LM}$. Single token criterion is to keep our experimental setup consistent. Lower bound criterion on test samples is to ensure average test CLIPScore is reliable for each target token. We report the macro average of test CLIPScores and overlap over the 30 extra target tokens in tables below. We'll add the detailed results in appendix for all baselines. We obtain results consistent with those in main paper, with Semi-NMF extracting concepts with good balance between high CLIPScore and low overlap.
>
> | Metric (Macro Avg)| Rnd-Words | Noise-Imgs | Simple | Semi-NMF |GT-captions|
> |:---:|:--:|:---:|:---:|:---:|:---:|
> | test CLIPScore $(\uparrow)$  | 0.515 | 0.521 | 0.625 | 0.630 | 0.762 |
>
>
> | Metric (Macro Avg) | Simple | PCA | KMeans | Semi-NMF |
> |:--:|:--:|:---:|:---:|:---:|
> | Overlap $(\downarrow)$  | 0.443 | 0.050 | 0.451 | 0.155 |
>
>
> **Application to multi-token words:** It is an interesting question. None of our evaluated tokens were in this category. However, our approach can be easily adapted to this. In particular, we extract representation of last token from first prediction of the multi-token sequence. The other aspects of the method remain unchanged. While there can also be other viable strategies, the rationale behind this adaptation is that the last token of our sequence of interest can also attend/combine representations from previous tokens in the sequence. We add below results for such examples. Detailed results will be added in appendix):
>
> |Metric|Token |Rnd-Words|Noise-Imgs|Simple|Semi-NMF|GT-captions|
> |:---:|:---:|:--:|:--:|:--:|:--:|:--:|
> |test CLIPScore $(\uparrow)$|traffic light|0.516 ± 0.03|0.525 ± 0.03|0.664 ± 0.06|0.634 ± 0.05|0.744 ± 0.04|
> |test CLIPScore $(\uparrow)$|cell phone|0.542 ± 0.04|0.547 ± 0.03|0.598 ± 0.04|0.598 ± 0.05|0.765 ± 0.06|
> |test CLIPScore $(\uparrow)$|stop sign|0.533 ± 0.03|0.549 ± 0.03|0.617 ± 0.08|0.616 ± 0.05|0.775 ± 0.04|
>
> | Metric | Token | Simple | PCA | K-Means | Semi-NMF |
> |:--:|:--:|:---:|:---:|:---:|:---:|
> | Overlap $(\downarrow)$|traffic light|0.704|0.050|0.579|0.174|
> | Overlap $(\downarrow)$|cell phone|0.623|0.051|0.746|0.164|
> | Overlap $(\downarrow)$|stop sign|0.461|0.058|0.704|0.109|
>
> **Motivation to use Semi-NMF:** The idea to consider Semi-NMF was inspired from the effectiveness of NMF for various interpretability applications ([1-4]). Given the positive and negative values in $h^p_{(L)}$, it was not possible to effectively apply NMF. Thus, Semi-NMF arose as a natural option by relaxing non-negativity on $\mathbf{U}$.  We expected the constraint to only positively combine the dictionary elements should still be useful from an interpretability perspective. Theoretical connections between Semi-NMF being generalized version of K-Means [5] also supported our confidence. We'll expand Sec 3.4 to include some of the motivation for more clarity. We'd also like to add that we did consider all three approaches (PCA, KMeans, Semi-NMF) initially and qualitatively found Semi-NMF dictionaries most balanced (diverse and multimodally meaningful).
>
>
> **Applicability of other current interpretability methods:** Related works (Sec. 2) discuss in detail the applicability of previous CAV-based approaches and previous approaches to understand VLMs/LMMs. Within general interpretability literature, the closest related methods to CAV based concept extraction are input autoencoder-based concept methods or concept bottleneck models. However these are used almost exclusively as by-design interpretable networks and thus out of scope for understanding pretrained models. We can add a brief discussion about them in Sec. 2.
>
>
> **Representations extraction + Relation to ROME:** We selected representation from first position where target token is predicted following Schwettmann et al. [6] who analyzed neurons for a given caption at the first predicted noun. In practice, what is essential is extracting representations where the target token is predicted. In this regard we are not in conflict with ROME but rather aligned. ROME analyzes factual associations of the form (subject, object, relation). The input prompt to the language model consists of the subject and relation. The model predicts the next token which is expected to be the object and thus ROME also analyzes representation at a position where their token of interest is predicted.
>
>
> **Repeated references:** Thanks. We'll correct them.
>
> [1] G. Trigeorgis et al. "A deep semi-nmf model for learning hidden representations." ICML 2014
>
> [2] J. Parekh et al. "Listen to interpret: Post-hoc interpretability for audio networks with nmf." NeurIPS 2022
>
> [3] T. Fel et al. "Craft: Concept recursive activation factorization for explainability." CVPR 2023
>
> [4] YT. Guo et al. "The rise of nonnegative matrix factorization: algorithms and applications." Information Systems 2024.
>
> [5] CHQ. Ding et al. "Convex and semi-nonnegative matrix factorizations." IEEE TPAMI 2008
>
> [6] S. Schwettmann et al. "Multimodal neurons in pretrained text-only transformers." ICCVW 2023

---

> > ### Comment · Reviewer_afLg · 2024-08-09
> >
> > Thank you for answering my questions and clarifying my doubts. I will keep my positive evaluation of the paper, and follow on the discussion with the other reviewers.

---

> > > ### Author Response · Authors · 2024-08-11
> > >
> > > Thanks for the rebuttal acknowledgement! We're glad to address your doubts.

---

### Official Review · Reviewer_XzMc · 2024-07-12

**Soundness:** 2
**Presentation:** 2
**Contribution:** 2
**Rating:** 4
**Confidence:** 2

**Summary:**

In this paper, the authors propose a framework for interpreting LMMs. Specifically, they introduce a dictionary learning-based approach applied to the representation of tokens. The elements of the learned dictionary correspond to the proposed concepts. These concepts are semantically well-grounded in both vision and text.

**Strengths:**

Using concepts to interpret large multimodal models is a promising idea.

**Weaknesses:**

1. I'm sorry, I don't fully understand this field, so I will lower my confidence score.
2. The notation system is quite confusing. I suggest the authors reorganize it for better clarity.
3. The paper directly applies the concept activation vector (CAV) to Large Multimodal Models, but it does not explain the benefits of doing so or why this approach is valid.

**Questions:**

The experiments in the paper are relatively few. I suggest conducting further research on the interpretability of multimodal large models such as LLaVA and MiniGPT-4.

**Limitations:**

yes, the author explains the limitations of their study and potential negative societal impact.

---

> ### Author Rebuttal · Authors · 2024-08-06
>
> We thank the reviewer for their feedback. We add our response for each point below:
>
> **Notation system:** We will try our best to reorganize some notations for clarity. We remain open to incorporate any particular suggestions.
>
>
> **Benefits of CAV-based approach for LMMs:** Concept based explainability approaches in general are preferred as they highlight what semantic representations a model extracts rather than which regions of input are important for a model [1]. In order to gain understanding about internal representations of an LMM, CAV-type approach were the only viable option for concept extraction as the other types of concept explainability methods (eg. Concept bottleneck models) focus on training interpretable networks by design. Furthermore, the effectiveness of recent works ([2]) in learning concepts dictionaries for CNNs also motivated us to develop a concept extraction approach that can be applied to LMMs.
>
> **New experiments:** We have conducted new experiments on LLaVA. We are able to extract meaningful multimodal concepts and achieve quantitatively consistent results as for DePALM. The global response and 1-page pdf present more details regarding the same.
>
> [1] J. Colin, T. Fel, R. Cadène, and T. Serre. "What I cannot predict, I do not understand: A human-centered evaluation framework for explainability methods." NeurIPS 2022.
>
> [2] T. Fel, V. Boutin, L. Béthune, R. Cadène, M. Moayeri, L. Andéol, M. Chalvidal, and T. Serre. "A holistic approach to unifying automatic concept extraction and concept importance estimation." NeurIPS 2023

---

> > ### Author Response · Authors · 2024-08-13
> >
> > Dear reviewer,
> >
> > We have incorporated your complete feedback in our rebuttal and sincerely hope it addresses all the concerns.

---

> > > ### Comment · Reviewer_XzMc · 2024-08-13
> > >
> > > I thank the authors for their endeavor in preparing the rebuttal. I have a new question: why are the results of MiniGPT4 missing in global response?

---

> > > > ### Author Response · Authors · 2024-08-13
> > > >
> > > > Thank you for the rebuttal acknowledgement. Due to the limited rebuttal time frame it was not possible for us to experiment with MiniGPT-4. We prioritized experiments on LLaVA because of its popularity and the competitiveness of its newer versions with state-of-the-art models like Gemini and GPT-4. Another point to note is that MiniGPT-4 has a similar architecture to LLaVA and thus there is no particular reason to expect its observations to be fundamentally different from LLaVA.
> > > >
> > > > In regard to experiments with other LMMs, we would also like to note our experiments in global rebuttal on other DePALM models with non CLIP visual encoders, where we obtain consistent results. We will certainly consider evaluating more LMMs, including MiniGPT4, for future works.
> > > >
> > > > We hope this clarifies your doubt.

---

### Official Review · Reviewer_Go37 · 2024-07-13

**Soundness:** 3
**Presentation:** 3
**Contribution:** 2
**Rating:** 4
**Confidence:** 4

**Summary:**

The authors propose using dictionary learning to extract concepts from multimodal models and simultaneously ground them in the text and image latent space. They draw on prior work on multimodal neurons and concept activation vectors. The authors provide quantitative results using CLIPScore and BERTScore to measure concept extraction, multimodal grounding, and concept overlap.

**Strengths:**

Appears to be mathematically correct and well-grounded in prior work.
Well-written and well-illustrated.

**Weaknesses:**

Evaluates only a few tokens (dog, bus, train, cat) thoroughly, which seems like not enough when prior work (Goh et al.) studies thousands of concepts.
Evaluates only one model, DePALM.
Uses two automated metrics, CLIPScore and BERTScore, but no human evaluation. A little worrisome that DePALM uses a CLIP-ViT-L14 and CLIPScore is the primary evaluation for the method.
Overlaps significantly with prior research (Goh et al, Kim et al); multimodal neurons have been described in work dating back several years using non-negative matrix factorization.

**Questions:**

In table 2, why is PCA by far the best method for minimizing overlap between learned concepts? Presumably because it’s PCA on only these five concepts, meaning that variance is maximized between only the five concepts and not between the dictionary more broadly?

Do you observe any evidence of polysemanticity, as reported in prior work including Goh et al.?

**Limitations:**

I don’t see many negative societal implications of this research.

---

> ### Author Rebuttal · Authors · 2024-08-06
>
> We thank the reviewer for their feedback. We address their concerns pointwise below:
>
> **Evaluates only on DePALM:** We have now also conducted experiments on LLaVA. We are able to extract meaningful multimodal concepts and achieve quantitatively consistent results as for DePALM. Further details regarding the same are available in the global response and 1-page PDF.
>
> **Uses two automated metrics, but no human evaluation:** Human evaluation would be a good way to further evaluate the performance of our approach. The current paper was rather focused on developing dictionary learning for LMMs, extracting multimodal concepts through dictionary learning, and studying various methods for this goal in a structured way. To the best of our knowledge, the feasibility of these aspects has not been explored before, and was one of the main concerns of the paper. Nevertheless, we would like to consider human evaluation in future developments.
>
> **Using CLIP as visual encoder and CLIPScore as metric:** We believe the meaningful grounding is influenced more by the language model. We conducted experiments with two non CLIP visual encoders to support this hypothesis. Please find further details in our global response.
>
> **Overlap/difference with prior research (Goh et al., Kim et al.) and multimodal neurons literature:**  We discuss both Goh et al., and TCAV (Kim et al.) in Sec. 2 along with our differences with them. Yes, the notion of multimodal neurons has been introduced previously. However, we respectfully disagree with the assessment that there is a significant overlap with it. Our approach performs dictionary learning on internal representations of a LMM, to extract a set of concepts about a target token. This methodology is quite unrelated to work in (Goh et al.) that analyzes activations of individual neurons in CLIP ResNet encoder (final conv layer) for various input images, to determine what conceptual information in image they activate for. In general, dictionary learning based concept extraction approaches and individual neuron analysis approaches are methodologically different but complementary. They are complementary approaches to analyze internal representations of a network. However, dictionary learning methods discover directions in the representation space useful to decompose data representations, while neuron analysis approaches study activation patterns of individual neurons in detail. We list some more significant differences with Goh et al.:
>
> - The ‘multimodality’ discussed in Goh et al. is different from ‘multimodality’ here or notion of multimodal neurons in Schwettmann et al. In particular, Goh et al. analyze neuron activations only for images where certain conceptual information can be present in different forms/modalities (eg. celebrity as a face in the image, as caricatures in image, as text in the image etc.), while the multimodality here or in Schwettman et al. concerns with explicit grounding of a concept vector or neuron in vision and text.
>
> - The two papers analyze very different types of models (LMMs vs CLIP-ResNet encoder) with different architectures, training objectives and tasks they solve.
>
> **Low overlap for PCA:** PCA is best for minimizing overlap possibly because its concept vectors by design are orthogonal. Thus, when decoded, they typically yield different sets of grounded words. We would also like to state for clarity that PCA is not learnt for just five concepts, nor do the target tokens ('dog', 'bus', 'cat', 'train') represent concepts of our dictionary. We apply PCA (or any dictionary learning method) to learn a different dictionary of $K=20$ concept vectors for each target token separately. Thus the overlap for dictionary learning methods is reported separately for each token and calculated between 20 concepts learnt for it.
>
> **Evidence of polysemanticity:** Our preliminary qualitative analysis suggests that concept vectors for prominent and distinct semantic concepts seem to be more monosemantic (eg. 'hot dog', 'black dog'). We also found concept vectors which are more polysemantic in nature, and capture more sparsely present semantic concepts. Such concepts might be identifiable in some cases while extracting the multimodal grounding. For instance, row 8, left column in Fig. 7 illustrates a concept discovered for 'Dog' token that activates for 'bull-dogs', 'pugs' but also for 'toy/stuffed dogs'. More details about the analysis can be found in our comment to Reviewer 17ab on feature superposition.

---

> > ### Author Response · Authors · 2024-08-13
> >
> > Dear reviewer,
> >
> > We have incorporated your complete feedback in our rebuttal and sincerely hope it addresses all the concerns.

---

### Official Review · Reviewer_8A1z · 2024-07-15

**Soundness:** 4
**Presentation:** 4
**Contribution:** 4
**Rating:** 8
**Confidence:** 4

**Summary:**

Authors propose to look at the representation of chosen concepts, in multimodal models. They test different automated methods to learn decompositions of a token representations (which they then linearise in a dictionary of concepts). They also provide quantitative and qualitative analysis of a few examples, showing that this method clusters different human-interpretable meaning or usages of a same concept, for both visual and textual modalities.

**Strengths:**

Great qualitative analysis
Multiple representation decomposition algorithms are tested, including more well known PCA or very specific but better suited Semi NMF, which is an adaptation to practical realities.
Evaluation of the method is quite extensive, using a wide range of relevant state of the art interpretability tools to verify their work theoretically and practically.
Effort is made to put quantitative metrics on concepts and to evaluate very abstract semantic analysis.
Provided examples are clear and strike curiosity.

**Weaknesses:**

I worry there is a circularity when evaluating with ClipScore a frozen model with a CLIP component.
(and even with BERT scoring, where you evaluate an interpretability metric with a non interpretable system of somewhat similar complexity).


(Minor)
Very recent works are relevant to the interpretability of LLM / Multimodal models (Templeton, et al., "Scaling Monosemanticity: Extracting Interpretable Features from Claude 3 Sonnet", Transformer Circuits Thread, 2024 AND Gao, Leo, et al. "Scaling and evaluating sparse autoencoders." arXiv preprint arXiv:2406.04093, 2024) and could be cited.

Line 223 "the correspondance between a the image" is probably a typo and should be fixed

**Questions:**

Mostly a personal curiosity question: Are there any intuitions from this work allowing to disentangle the data effect of the apparition of the studied concepts from the model/training effect?

**Limitations:**

Sample experiments are done on very simple words and concepts, available at pre-training for the tested model. How well this method would adapt to more complex concepts and images is not studied, but is where a lot of interpretability becomes necessary. This is nonetheless a necessary first step in an exciting direction.
Do you have an intuition on how well this method would scale to this?

---

> ### Author Rebuttal · Authors · 2024-08-06
>
> We thank the reviewer for their positive comments and intriguing questions. We respond to them pointwise below:
>
> **CLIPScore and LMM with frozen CLIP encoder:** We believe the meaningful grounding is influenced more by the language model. We conducted experiments with two non CLIP visual encoders to support this hypothesis. Please find further details in our global response.
>
> **Very recent works on LLM interpretability:** Thanks, we make a note of them.
>
> **Line 223 typo:** Thanks for pointing out the error. We’ll fix it.
>
> **Disentangling data effect/model effect:** Indeed, the question of the impact of the available data vs the model architecture/training, w.r.t the concepts, is interesting. Qualitatively, in our initial experiments we observed cases where the type of input data clearly affected the quality of extracted concepts. For instance, when extracting concepts for token 'light', we observed most concepts only contained images of 'traffic light', even though we expected more diversity. This was because most input images in our dataset with predicted token 'light' were of 'traffic light'.
>
> On a separate note, it could be worth tracking the evolution of the learnt concepts of our approach during training. This could possibly help to disentangle the training/optimization effect. But we leave that as a future research direction to pursue.
>
> **Expanding to more complex concepts:** Yes, this is indeed an important direction we are considering to expand in future. One possible way to extend to more complex or abstract words could be to associate the complex word to a set of target tokens. We could then consider decomposing representations for the all tokens in the set simultaneously. Our first instinct is to use a linear decomposition approach. Nevertheless, if this is not sufficient, non-linear decomposition methods could also be explored.

---

> > ### Comment · Reviewer_8A1z · 2024-08-09
> >
> > Thank you for your response. I will maintain  my score, as I continue to believe it is an interesting and well developed paper, and will follow discussion with other reviewers.

---

> > > ### Author Response · Authors · 2024-08-11
> > >
> > > Thank you for the rebuttal acknowledgement!

---

### Author Rebuttal · Authors · 2024-08-06

We want to thank all the reviewers for their great interest and useful feedback. We address most reviewer comments individually. In this global post we would like to address two key concerns, each raised by at least two reviewers:

1. **New experiments on LLaVA (Reviewers Go37, XzMc, 17ab):**  We have now conducted experiments on LLaVA-v1.5 model. The model uses a CLIP-ViT-L-336px visual encoder, a 2-layer linear mapper that outputs $N_V=576$ visual tokens, and a Vicuna-7B language model (32 layers). We use identical hyperparameters as for DePALM ($K=20, \lambda=1, L=31$). The **attached 1-page PDF** contains the figures/tables regarding this experiment. For now, we report (i) the test CLIPScore for top-1 activating concept, for Rnd-Words, Noise-Imgs, Simple and Semi-NMF, GT-captions (Tab. 1 in 1-page PDF) and (ii) Overlap score for non-random baselines (Tab. 2 in 1-page PDF). We also show qualitative examples of concepts extracted for token `Dog' (Fig. 1 in 1-page PDF). More detailed quantitative and qualitative results will be added in appendix. Quantitatively, we obtain **consistent results** to those observed for DePALM. Semi-NMF extracts most balanced concept dictionaries with high multimodal correspondence and low overlap. Qualitatively too, the method functions consistently and is able to extract concepts with meaningful multimodal grounding.

2. **CLIP visual encoder and CLIPScore as metric (Reviewers 8A1z, Go37):** The meaningful concept representations, we believe, are more due to their alignment with language model representations than the visual encoder. To support our hypothesis, we conducted experiments with 2 different DePALM models with frozen visual encoders different from CLIP, a frozen ViT-L encoder trained on ImageNet [1] and another frozen ViT-L trained as a masked autoencoder (MAE) [2]. Both LMMs use the same pretrained OPT-6.7B language model. Collectively, the three encoders (including CLIP) are pretrained for three different types of objectives. We use Semi-NMF to extract concept dictionaries, with all hyperparameters identical. The results are reported in tables below. 'Rnd-Words' and 'GT-captions' references are reported for each LMM separately, although they are very close to the ones in main paper. The "ViT-L (CLIP)" baseline denotes our system from the main paper that uses CLIP encoder. Importantly, we still obtain similar test CLIPScores as with CLIP visual encoder. The concept dictionaries still possess meaningful multimodal grounding. The various concepts also tend to be similar as for CLIP visual encoder. We'll add the qualitative visualizations and complete quantitative results for the non-CLIP encoders in appendix.

| Token         | Rnd-Words  | ViT-L (ImageNet)  | ViT-L (CLIP)  | GT-captions |
|:-------------:|:----------:|:---------------:|:--------------:|:-----------:|
| Dog           | 0.514 ± 0.05 | 0.611 ± 0.09 | 0.610 ± 0.09 | 0.783 ± 0.06 |
| Bus           | 0.498 ± 0.05 | 0.644 ± 0.07 | 0.634 ± 0.08 | 0.739 ± 0.05 |
| Train         | 0.494 ± 0.05 | 0.617 ± 0.07 | 0.646 ± 0.07 | 0.728 ± 0.05 |
| Cat           | 0.539 ± 0.05 | 0.628 ± 0.07 | 0.627 ± 0.06 | 0.794 ± 0.06 |

| Token         | Rnd-Words  | ViT-L (MAE)  | ViT-L (CLIP)  | GT-captions |
|:-------------:|:----------:|:--------------:|:------------:|:-----------:|
| Dog           | 0.515 ± 0.05 | 0.602 ± 0.07 | 0.610 ± 0.09 | 0.784 ± 0.06 |
| Bus           | 0.501 ± 0.05 | 0.627 ± 0.07 | 0.634 ± 0.08 | 0.737 ± 0.05 |
| Train         | 0.483 ± 0.06 | 0.618 ± 0.08 | 0.646 ± 0.07 | 0.726 ± 0.05 |
| Cat           | 0.541 ± 0.04 | 0.629 ± 0.09 | 0.627 ± 0.06 | 0.795 ± 0.06 |

[1] A. Dosovitskiy et al. "An image is worth 16x16 words: Transformers for image recognition at scale." ICLR 2021

[2] K. He, X. Chen, S. Xie, Y. Li, P. Dollár, R. Girshick. "Masked autoencoders are scalable vision learners." CVPR 2022

---

### Decision · Program_Chairs · 2024-09-25

**Decision:**

Accept (poster)

**Comment:**

This paper proposes a dictionary-learning approach to analyzing multimodal LMs.  A qualitative/quantitative analysis shows the emergence of clearly interpretable multi-modal concepts.

There was some discrepancy in the reviewers' opinion about the paper, and some important concerns were voiced, pertaining to the limited scope of the evaluation, the fact that there is no subject-based manual validation of the results, and the general issue of the danger of interpreting concepts in a monosemic way when representation superposition is a widespread phenomenon in neural networks. However, the method is clever and leads to promising results, and it could generally help people working on LM interpretability. Moreover, the reviewers that provided the more detailed reviews and more actively engaged with the authors were also those that have a more positive opinion of the paper. I would tend to agree with them, and I would like to see this work at the conference.